# SWARM Parallelism: Training Large Models Can Be Surprisingly Communication-Efficient

## Abstract

Many deep learning applications benefit from using large models with billions of parameters. Training these models is notoriously expensive due to the need for specialized HPC clusters. In this work, we consider alternative setups for training large models: using cheap "preemptible" instances or pooling existing resources from multiple regions. We analyze the performance of existing model-parallel algorithms in these conditions and find configurations where *training larger models becomes less communication-intensive*. Based on these findings, we propose SWARM parallelism[1], a model-parallel training algorithm designed for poorly connected, heterogeneous and unreliable devices. SWARM creates temporary randomized pipelines between nodes that are rebalanced in case of failure. We empirically validate our findings and compare SWARM parallelism with existing large-scale training approaches. Finally, we combine our insights with compression strategies to train a large Transformer language model with 1B shared parameters ($\approx$13B before sharing) on preemptible T4 GPUs with less than 200Mb/s network.

## 1 Introduction

For the past several years, the deep learning community has been growing ever more reliant on large pretrained neural networks. Perhaps the easiest example of this trend is natural language processing, where the parameter count of models grew from hundreds of millions (Vaswani et al., 2017; Radford et al., 2018; Devlin et al., 2019) to billions (Narayanan et al., 2021; Rosset; Raffel et al., 2020; Wang & Komatsuzaki, 2021; Sun et al., 2021) to hundreds of billions (Brown et al., 2020; Lepikhin et al., 2020; Fedus et al., 2021; Chowdhery et al., 2022; Rae et al., 2021) with consistent gains in quality (Kaplan et al., 2020). Likewise, many models in computer vision are reaching the billion-parameter scale (Henighan et al., 2020; Ramesh et al., 2021; Zhai et al., 2021; Riquelme et al., 2021; Dai et al., 2021; Dhariwal & Nichol, 2021).

At this scale, the models no longer fit into a single accelerator and require specialized training algorithms that partition the parameters across devices (Krizhevsky et al., 2012; Dean et al., 2012). While these model-parallel algorithms use different partitioning strategies, they all share the need to perform intensive device-to-device communication (Narayanan et al., 2019; 2021). Furthermore, if a single device fails, it will cause the entire training process to break down. As a result, model-parallel algorithms are typically deployed in dedicated high-performance computing (HPC) clusters or supercomputers (Shoeybi et al., 2019; Rajbhandari et al., 2020; Narayanan et al., 2021).

This kind of infrastructure is notoriously expensive to build and operate, available only to a few well-funded universities and large corporations (Larrea et al., 2019; Strohmaier et al., 2021; Langston, 2020). Most researchers, especially in developing nations, cannot afford the experiments necessary for a proper evaluation of their ideas. This ultimately limits the scientific progress for many important research areas, such as solving NLP problems in "non-mainstream" languages.

Several recent works propose more cost-efficient distributed training strategies leveraging fleets of temporary "preemptible" instances that can be dynamically allocated in regions with low demand for hardware and electricity, making them 2–10 times cheaper than their dedicated counterparts (Harlap et al., 2017). Another solution is to train in "collaborations" by pooling together preexisting resources or using the help of volunteers (Diskin et al., 2021; Atre et al., 2021; Ryabinin & Gusev, 2020).

---

[1]SWARM parallelism is a backronym for Stochastically Wired Adaptively Rebalanced Model Parallelism.

However, training in either of those setups requires specialized algorithms that can adapt to the changing number of workers, utilize heterogeneous devices and recover from hardware and network failures. While there are several practical algorithms for unreliable hardware (Kijsipongse et al., 2018; Lin et al., 2020; Ryabinin et al., 2021), they can only train relatively small models that *fit into the memory of the smallest device*. This limits the practical impact of cost-efficient strategies, as most computationally demanding workloads typically train models with billions of parameters.

In this work, we aim to find a practical way of training large neural networks using **unreliable heterogeneous devices and slow interconnect**. We begin by studying the impact of model size on the balance between communication and computation costs of pipeline-parallel training. Specifically, increasing the size leads computation costs to grow faster, thus rendering the bottleneck of Internet-grade network speeds negligible. This idea inspires the creation of SWARM parallelism — a pipeline-parallel approach designed to handle peer failures by using randomized routing that prioritizes stable peers with lower latency. In addition, this approach periodically rebalances the pipeline stages, which allows handling devices with different hardware and network speeds.

In summary, we make the following contributions:

- We carefully analyze the existing model-parallel training techniques and formulate the "Square-Cube Law" of distributed training: a counterintuitive observation that, for some methods, *training larger models can actually decrease the network overhead*.
- We develop SWARM parallelism, a decentralized model-parallel algorithm[2] that leverages randomized fault-tolerant pipelines and dynamically rebalances nodes between pipeline stages. To the best of our knowledge, this is the first algorithm capable of billion-scale training on heterogeneous unreliable devices with slow interconnect.
- Combining insights from the square-cube law, SWARM parallelism, and 8-bit compression, we show that it is possible to train a billion-scale Transformer language model with high throughput on preemptible low-power T4 GPUs with $< 200$Mb/s network bandwidth.

## 2 BACKGROUND & RELATED WORK

### 2.1 MODEL-PARALLEL TRAINING

Over the past decade, the deep learning community has developed several algorithms for training large neural networks. Most of them work by dividing the model between multiple workers, which is known as model parallelism. The exact way in which these algorithms divide the model determines their training performance and the maximum model size they can support.

**Traditional model parallelism.** Historically, the first general strategy for training large models was to assign each device to compute a subset of each layer (e.g., a subset of neurons), then communicate the results between each other (Krizhevsky et al., 2012; Ben-Nun & Hoefler, 2019; Tang et al., 2020). Since each device stores a fraction of layer parameters, this technique can train models with extremely wide layers that would not fit into a single GPU. However, applying traditional model parallelism to deep neural networks comes at a significant performance penalty, as it requires all-to-all communication after each layer. As a result, while intra-layer parallelism is still widely used (Shazeer et al., 2018; Rajbhandari et al., 2020), it is usually applied within one physical server in combination with other strategies (Krizhevsky, 2014; Chilimbi et al., 2014; Jia et al., 2019; Narayanan et al., 2021).

**Pipeline parallelism** circumvents the need for expensive all-to-all communication by assigning each device with one or several layers (Huang et al., 2019). During the forward pass, each stage applies its subset of layers to the inputs supplied by the previous stage, then sends the outputs of the last layer to the next stage. For the backward pass, this process is reversed, with each pipeline stage passing the gradients to the same device that previously supplied it with input activations.

To better utilize the available devices, the pipeline must process multiple microbatches per step, allowing each stage to run in parallel on a different batch of inputs. In practice, the number of microbatches is limited by the device memory: this results in reduced device utilization when

---

[2] The code for our experiments can be found at `github.com/iclr2023-submit/swarm`

processing the first and the last microbatches, known as the "bubble" overhead (Huang et al., 2019). To combat this issue, subsequent studies propose using activation checkpointing, interleaved scheduling, and even asynchronous training (Narayanan et al., 2019; 2021; Huang et al., 2019; Shoeybi et al., 2019; Yang et al., 2019).

Aside from model parallelism, there two more strategies for training large models: data parallelism with dynamic parameter loading (Rajbhandari et al., 2020) and model-specific algorithms such as Mixture-of-Experts (Shazeer et al., 2017). We discuss these algorithms in Appendix B and compare the performance of offloading with SWARM in Section 4.2 and Appendix E.

## 2.2 DISTRIBUTED TRAINING OUTSIDE HPC

The techniques described in Section 2.1 are designed for clusters of identical devices with rapid and reliable communication, making them a natural fit for the HPC setup. As we discussed earlier, such infrastructure is not always available, and a more cost-efficient alternative is to use "preemptible" instances (Li et al., 2019; Zhang et al., 2020; Harlap et al., 2017) or volunteer computing (Kijsipongse et al., 2018; Ryabinin & Gusev, 2020; Atre et al., 2021; Diskin et al., 2021). However, these environments are more difficult for distributed training: each machine can disconnect abruptly due to a failure or preemption. Besides, since there is a limited number of available instances per region, training at scale often requires operating across multiple locations or using different instance types.

To handle unstable peers and heterogeneous devices, the research community has proposed elastic training and asynchronous distributed methods, correspondingly. We describe these approaches in more detail in Appendix B; most importantly, they rely on data-parallel training, and thus each node must be able to run the entire model.

By contrast, the largest models have billions of parameters, which exceeds the memory limits of most low-end computers. However, model-parallel algorithms are not redundant, which makes them more vulnerable to hardware and network failures. As far as we know, there are two methods that allow training large models with unreliable devices (Ryabinin & Gusev, 2020; Thorpe et al., 2022): however, the first one supports only specific architectures and requires at least 1Gb/s bandwidth, whereas the second one has no publicly available implementations, relies on redundant computations for fault tolerance and considers only the homogeneous setup.

## 2.3 COMMUNICATION EFFICIENCY AND COMPRESSION

In this section, we discuss techniques that address training with limited network bandwidth or high latency, such as gradient compression or overlapping computation with communication phases. These techniques are often necessary for distributed training without high-speed connectivity, because otherwise the performance of the system becomes severely bottlenecked by communication.

**Efficient gradient communication.** Data-parallel training requires synchronization of gradients after each backward pass, which can be costly if the model has many parameters or the network bandwidth is limited. Deep Gradient Compression (Lin et al., 2018) approaches this problem by sparsifying gradients before synchronizing and correcting the momentum to work with sparse updates. PowerSGD (Vogels et al., 2019) compresses gradients via factorization and uses error feedback to compensate the approximation errors over time. Dettmers (2015) uses nonlinear 8-bit quantization to compress gradients before communication. We evaluate this method along with compression-aware architectures and leave the exploration of more advanced approaches to future work.

Besides gradient compression, another effective technique is to use layer sharing (Lan et al., 2020), which reduces the number of aggregated gradients by a factor of how many times each layer is reused.

**Overlapping communication and computation.** Model, pipeline, and data parallelism all have synchronization points and require transfer of gradients or activations. One way to reduce the transfer cost is to overlap communication with computation, *hiding* the synchronization latency. This overlap can be achieved by combining parallelization techniques (Krizhevsky, 2014; Rajbhandari et al., 2020), by synchronizing gradients layer-by-layer in lockstep with backpropagation (Paszke et al., 2019), or by using pure pipeline parallelism (Huang et al., 2019; Narayanan et al., 2019). However, pure pipeline parallelism requires many stages to effectively hide the latency. To overcome this problem, we study inter-layer compression techniques that work well even with relatively few pipeline stages.

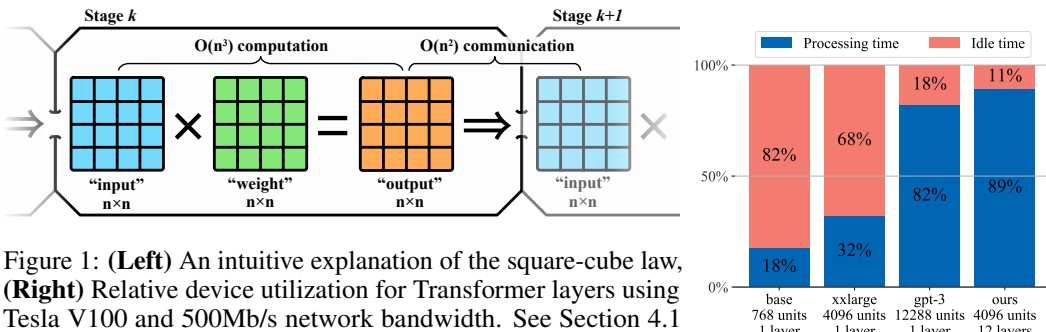

Figure 1: **(Left)** An intuitive explanation of the square-cube law, **(Right)** Relative device utilization for Transformer layers using Tesla V100 and 500Mb/s network bandwidth. See Section 4.1 and Appendix F for a detailed experimental setup.

## 3 COMMUNICATION-EFFICIENT MODEL PARALLELISM

In this section, we outline our approach for training large models with heterogeneous unreliable poorly-connected devices. To that end, the section is organized as follows:

- Section 3.1 analyzes how existing model-parallel algorithms scale with model size and find conditions where increasing training large models leads to less intense network usage;
- Section 3.2 describes SWARM parallelism — a decentralized algorithm for training large models under the challenges outlined in Section 2.2.

### 3.1 THE SQUARE-CUBE LAW OF DEEP LEARNING

To better understand the general scaling properties of model parallelism, we must abstract away from the application-specific parameters, such as model architecture, minibatch size, and system design. To that end, we first consider a simplified model of pipeline parallelism. Our "pipeline" consists of $k$ stages, each represented by $n \times n$ matrices. Intuitively, the first matrix represents input data and all subsequent matrices are linear "layers", applied to that data. This model abstracts away from application-specific details, allowing us to capture general relationships that hold for many models.

During "training", stages iteratively perform matrix multiplication and then send the output to the subsequent pipeline stage over a throughput-limited network. These two operations have different scaling properties. The compute time for naïve matrix multiplication scales as $O(n^3)$. While this can be reduced further in theory (Coppersmith & Winograd, 1990; Alman & Williams, 2021), it is only used for very large matrices (Zhang & Gao, 2015; Fatahalian et al., 2004; Huang et al., 2020). Therefore, deep learning on GPUs typically relies on $O(n^3)$ algorithms.

In turn, the communication phase requires at most $O(n^2)$ time to transfer a batch of $n \times n$ activations or gradients. Therefore, as we increase the model size, the computation time grows faster than communication time, regardless of which matrix multiplication algorithm we use. We refer to this idea as the *square-cube law* after the eponymous principle in physics (Galileo, 1638; Allen, 2013).

This principle applies to many real-world neural network architectures, albeit with some confounding variables. In convolutional neural networks Fukushima (1980), the computation time scales as $O(BHWC^2)$ and the communication is $O(BHWC)$, where $B$, $H$, $W$ and $C$ stand for batch size, height, width and the number of channels. Recurrent neural networks (Rumelhart et al., 1986; Hochreiter & Schmidhuber, 1995) need $O(BLH^2)$ compute in terms of batch size, sequence length, and hidden size, respectively, and $O(BLH)$ or $O(BH)$ communication, depending on the architecture. With the same notation, Transformers (Vaswani et al., 2017) require $O(BL^2H)$ compute for attention layers, $O(BLH^2)$ compute for feedforward layers, but only $O(BLH)$ communication.

Based on these observations, we conclude that pipeline parallelism naturally grows more communication-efficient with model size. More precisely, increasing the hidden dimension will reduce the communication load per device per unit of time, making it possible to train the model efficiently *with lower network bandwidth* and *higher latency*[3]. While the exact practical ramifications depend on the use case, Section 4.1 demonstrates that some of the larger models trained with pipeline parallelism can already train at peak efficiency with only hundreds of Mb/s bandwidth.

---

[3]Latency slows the communication down by a constant factor that also grows less important with model size.

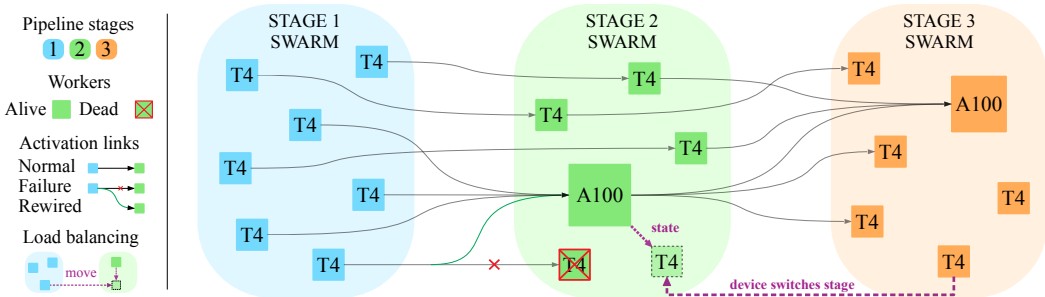

Figure 2: An overview of SWARM parallelism, illustrating both normal operation, device failures and adaptive rebalancing. One of the workers at stage 2 leaves; another peer from stage 3 takes its place by downloading the latest stage 2 parameters and statistics from peers.

In theory, the square-cube principle also applies to intra-layer parallelism, but using this technique at 500 Mb/s would become practical only for layer sizes of more than $2^{16}$ units. Data-parallel training with sharding or offloading (Ren et al., 2021) does not scale as well, as its communication time scales with the size of *model parameters* instead of activations. However, it may be possible to achieve similar scaling with gradient compression algorithms.

## 3.2 SWARM PARALLELISM

Traditional pipeline parallelism can be communication-efficient, but this alone is not enough for our setups. Since training devices can have different compute and network capabilities, a pipeline formed out of such devices would be bottlenecked by the single "weakest link", i.e., the participant with the smallest training throughput. As a result, the more powerful nodes along the pipeline would be underutilized due to either lack of inputs or slow subsequent stages. On top of that, if any node fails or leaves training prematurely, it will stall the entire training procedure.

In order to overcome these two challenges, we replace the rigid pipeline structure with randomized temporary "pipelines" that are wired stochastically on the fly during each iteration. Each participant can send their outputs to any peer that serves the next pipeline stage. Thus, if one peer has significantly more compute than others, it can process inputs from multiple predecessors and distribute its outputs across several weaker peers to maximize utilization. Furthermore, if a participant leaves during training, its predecessors can reroute their requests to its neighbors, and joining peers can download up-to-date parameters and optimizer statistics from remaining workers at the chosen stage. This allows the training to proceed as long as there is at least one active participant per pipeline stage (we elaborate on fault tolerance of SWARM in Appendix A).

The resulting system consists of several consecutive swarms, as depicted in Figure 2. Peers within one swarm serve the same pipeline stage (i.e., the same subset of layers with the same parameters). We assume that the model consists of similar "blocks" and thus partition it into evenly sized stages; we leave the study of better strategies (Huang et al., 2019; Narayanan et al., 2019) for our setting to future work. During the *forward* pass, peers receive microbatches of inputs from random predecessors (determined on each iteration) and send activations to random peers in the next stage. For the *backward* pass, peers receive gradients for outputs, compute gradients for layer inputs and accumulate gradients for parameters. Once enough gradients are accumulated, peers form groups, run All-Reduce to average gradients within their respective pipeline stages and run a global optimizer step.

SWARM can also use Delayed Parameter Updates (DPU) (Ren et al., 2021) to further improve device utilization by performing the optimizer step in parallel with processing the next batch. While it is technically asynchronous, DPU was shown to achieve similar per-iteration convergence as fully synchronous training, both theoretically (Stich & Karimireddy, 2020; Arjevani et al., 2020) and empirically (Ren et al., 2021; Diskin et al., 2021).

Each peer has queues for incoming and outgoing requests to maintain high GPU utilization under latency and to compensate for varying network speeds. Similarly to other pipeline implementations (Huang et al., 2019; Narayanan et al., 2021), SWARM uses activation checkpointing (Griewank & Walther, 2000; Chen et al., 2016) to reduce the memory footprint.

**Stochastic wiring.** To better utilize heterogeneous devices and recover from faults, we dynamically "wire" each input through each stage and pick devices in proportion to their training throughput. To achieve this, SWARM peers run "trainer" processes that route training data through the "stages" of SWARM, balancing the load between peers.

For each pipeline stage, trainers discover which peers currently serve this stage via a Distributed Hash Table (DHT, Maymounkov & Mazieres, 2002). Trainers then assign microbatch to one of those peers in proportion to their performance. If that peer fails, it is temporarily banned and the microbatch is sent to another peer within the same stage. Note that trainers themselves do not use GPUs and have no trainable parameters, making it possible to run multiple trainers per peer.

Each trainer assigns data independently using Interleaved Weighted Round-Robin (Katevenis et al., 1991; Tabatabaee et al., 2020) scheduler. Our specific implementation of IWRR uses a priority queue: each peer is associated with *the total processing time over all previous requests*. A training minibatch is then routed to the node that has the smallest total processing time. Thus, for instance, if device A takes half as long to process a sample as device B, the routing algorithm will choose A twice as often as B. Finally, if a peer does not respond or otherwise fails to process the minibatch, trainer will "ban" this peer until it reannounces itself, which is done every few minutes.

Curiously, different trainers can have different throughput estimates for the same device based on the network topology. For instance, if training nodes are split between two cloud regions, a given peer's trainer will have a higher throughput estimate for peers in the same data center. In other words, trainers automatically adjust to the network topology by routing more traffic to peers that are "nearby". For a more detailed description of stochastic wiring, please refer to Appendix C.

**Adaptive swarm rebalancing.** While stochastic wiring allows for automatic rebalancing within a stage, additional cross-stage rebalancing may be required to maximize throughput, especially when devices are very unreliable. As we described earlier in Section 2.2, our workers can join and leave training at any time. If any single pipeline stage loses too many peers, the remaining ones will have to deal with an increased processing load, which will inevitably form a bottleneck.

SWARM parallelism addresses this problem by allowing peers to dynamically switch between "pipeline stages" to maximize the training throughput. Every $T$ seconds, peers measure the utilization rate of each pipeline stage as the queue size. Peers from the most underutilized pipeline stage will then switch to the most overutilized one (see Figure 2 for an overview and Appendix D for a formal description and complexity analysis), download the latest training state from their new neighbors and continue training. Similarly, if a new peer joins midway through training, it is assigned to the optimal pipeline stage by following the same protocol. As a side effect, if one pipeline stage requires more compute than others, SWARM will allocate more peers to that stage (see Appendix H).

## 4 EXPERIMENTS

### 4.1 COMMUNICATION EFFICIENCY AT SCALE

Before we can meaningfully evaluate SWARM parallelism, we must verify our theoretical observations on communication efficiency. Here we run several controlled experiments that measure the GPU utilization and network usage for different model sizes, using the Transformer architecture (Vaswani et al., 2017) that has been widely adopted in various fields (Lin et al., 2021). To decouple the performance impact from other factors, we run these experiments on homogeneous V100 GPU nodes that serve one pipeline stage over the network with varying latency and bandwidth. We use a batch size of 1 and sequences of 512 tokens; the complete configuration is deferred to Appendix F.

First, we measure how the model size affects the computation to communication ratio at 500 Mb/s network bandwidth in both directions. We consider 4 model configurations: the base configuration from the BERT paper (Devlin et al., 2019), "xxlarge" ("large" with $d_{model}$=4096), which is used in several recent works (Lan et al., 2020; Sun et al., 2021; He et al., 2020), and a GPT-3-scale model with $d_{model}$=12288 (Brown et al., 2020). We also evaluate a modified Transformer architecture ("Ours") as defined in Section 4.3 with $d_{model}$=4096, 3 layers per pipeline stage and 8-bit quantized activations. As we demonstrate in Appendix K, this compression strategy can significantly reduce network usage with little effect on convergence. In the first three configurations, the model consists of 12 Transformer layers placed on 12 servers with a single GPU; in the last one, there are 4 servers, each hosting 3 layers. Appendix F contains FLOP and parameter counts of each configuration.

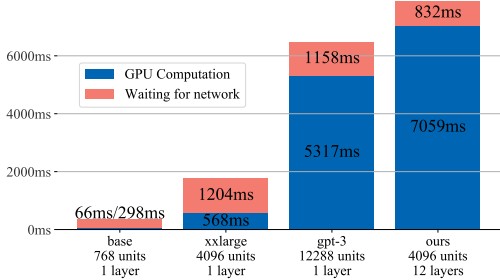

Figure 3: Pipeline computation and idle time per batch at 500 Mb/s bandwidth.

Table 1: Relative device utilization at 500 Mb/s bandwidth and varying network latency.

| Latency (RTT) | Relative GPU utilization (100% - idle time) | | | |
|---|---|---|---|---|
| | base | xxlarge | GPT-3 | Ours |
| none | 18.0% | 32.1% | 82.1% | 89.5% |
| 10ms | 11.8% | 28.9% | 79.3% | 87.2% |
| 50ms | 4.88% | 20.1% | 70.3% | 79.5% |
| 100ms | 2.78% | 14.9% | 60.2% | 71.5% |
| 200ms | 1.53% | 10.1% | 48.5% | 59.2% |

As depicted in Figure 1 (right) and Figure 3, larger models achieve better GPU utilization rate in the same network conditions, since their communication load grows slower than computation. More importantly, even at 500 Mb/s, the resulting GPU idle time can be pushed into the 10–20% range, either naturally for GPT-3-sized models or through activation compression for smaller models. In addition, large models maintain most of their training efficiency at the 100ms latency (Table 1), which is roughly equivalent to training on different continents (Verizon, 2021).

## 4.2 Detailed performance comparison

Here we investigate how SWARM parallelism compares to existing systems for training large models: **GPipe** (Huang et al., 2019) and **ZeRO-Offload** (Ren et al., 2021). The purpose of this section is to compare the training throughput in "ideal" conditions (with homogeneous reliable devices and balanced layers), as deviating from these conditions makes it *infeasible* to train with baseline systems. We benchmark individual SWARM components in preemptible setups in Appendices H and I.

We evaluate training performance for sequences of 4 Transformer layers of identical size distributed over 16 workers. The pipeline does not contain embeddings or language modeling heads, as it would result in imbalance between the stages. Similarly to Section 4.1, we use two layer configurations: "xxlarge" ($d_{model}$=4096, $d_{FFN}$=16384, 32 heads) and "GPT-3" ($d_{model}$=12288, $d_{FFN}$=49152, 96 heads). The microbatch size is 4 for "xxlarge" and 1 for "GPT-3", and the sequence length is 512.

To provide a more detailed view of the training performance, we measure two separate performance statistics: the training throughput and the All-Reduce time. The training throughput measures the rate at which the system can process training sequences, i.e., run forward and backward passes. In turn, the All-Reduce time is the time each system spends to aggregate those accumulated gradients across devices. The total time per step can be computed as `batch_size / throughput + all_reduce_time`. Intuitively, training with small batch sizes is more sensitive to the All-Reduce time (since the algorithm needs to run All-Reduce more frequently) and vice versa.

**Hardware setup:** Each worker uses a V100-PCIe GPU with 16 CPU threads (E5 v5-2660v4) and 128 GB RAM. The only exception is for ZeRO-Offload with "GPT-3" layers, where we had to double the RAM size because the system required 190GB at peak. Similarly to Section 4.1, each worker can communicate at a 500 Mb/s bandwidth for both upload and download for a total of 1 Gb/s. In terms of network latency, we consider two setups: with **no latency**, where workers communicate normally within the same rack, and with **latency**, where we inject additional $100 \pm 50$ms latency in the kernel[4].

**GPipe configuration:** We use a popular PyTorch-based implementation of GPipe[5]. The model is partitioned into 4 stages repeated over 4 model-parallel groups. To fit into the GPU memory for the "GPT-3" configuration, we offload the optimizer into RAM using ZeRO-Offload. Before averaging, we use PyTorch's built-in All-Reduce to aggregate gradients. We evaluate both the standard GPipe schedule and the 1F1B (Narayanan et al., 2019) schedule.

**ZeRO-Offload configuration:** Each worker runs the entire model individually, then exchanges gradients with peers. For "xxlarge", we use the official implementation from Ren et al. (2021). However, for "GPT-3", we found that optimizer offloading still does not allow us to fit 4 layers into the GPU. For this reason, we also offload the model parameters using the `offload_param` option.

---

[4]More specifically, `tc qdisc add dev <...> root netem delay 100ms 50ms`

[5]The source code is available at `https://github.com/kakaobrain/torchgpipe`

Table 2: Training performance for different model sizes.

| System | Throughput, samples/s | | All-Reduce time, s/round | |
|---|---|---|---|---|
| | No latency | Latency | No latency | Latency |
| "GPT-3" | | | | |
| SWARM | 0.619 | **0.558** | 441.7 | **455.4** |
| GPipe | 0.633 | 0.477 | **403** | 469.6 |
| 1F1B | **0.638** | 0.482 | | |
| Offload | 0.382 | 0.382 | 1527.9 | 1635.4 |
| "xxlarge" | | | | |
| SWARM | 2.358 | 2.161 | 45.36 | **51.269** |
| GPipe | 2.541 | 0.957 | **44.17** | 64.828 |
| 1F1B | 2.550 | 0.987 | | |
| Offload | **3.08** | **3.08** | 168.71 | 252.26 |

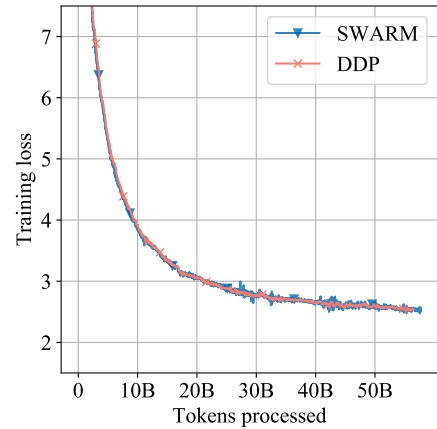

Figure 4: Training convergence comparison.

In turn, when training smaller models, ZeRO-Offload outperforms both SWARM and GPipe. This result aligns with our earlier observations in Figure 1, where the same model spent most of the time waiting for the communication between pipeline stages.

We also observe that ZeRO-Offload takes longer to aggregate gradients, likely because each peer must aggregate the entire model, whereas in SWARM and GPipe, peers aggregate a single pipeline stage. The variation between All-Reduce time in GPipe and SWARM is due to implementation differences. Overall, SWARM is competitive to HPC baselines even in an idealized homogeneous environment.

### 4.3 LARGE-SCALE DISTRIBUTED TRAINING

Finally, to verify the efficiency of SWARM parallelism in a practical scenario, we conduct a series of large-scale distributed experiments using preemptible (unreliable) cloud T4 and A100 GPUs over a public cloud network. In all experiments below, we train a Transformer language model similar to prior work (Brown et al., 2020; Wang & Komatsuzaki, 2021; Black et al., 2021) with 1.01 billion parameters in total: because of layer sharing, it is equivalent to a 13B model from (Brown et al., 2020) in terms of compute. Our model consists of 3 stages, each containing a single Transformer decoder block with $d_{model} = 4096$ and 16 layers per pipeline stage. We use 8-bit compression (Dettmers et al., 2021) for activations and gradients to reduce the communication intensity.

All workers within each stage serve the same group of layers independently of other stages, and all layers within each group use the same set of parameters, similarly to the ALBERT model (Lan et al., 2020). On top of this, the first stage also contains the embedding layer, and the last stage includes the language modeling head. Additional training setup details are covered in Appendix G. SWARM nodes perform rebalancing every $T = 300$ seconds, and trainers measure device performance using a moving average with $\alpha = 0.1$. However, as we show in Appendix H, the throughput of SWARM is not very sensitive to the choice of these hyperparameters.

First, to verify that model parallelism with asynchronous updates does not have significant convergence issues, we train the model on the Pile (Gao et al., 2020) dataset with 400 preemptible T4 instances, each hosting one accelerator. As a baseline, we use regular data-parallel training with offloading on 128 A100 GPUs. We run both experiments for approximately 4 weeks and compare the learning curves.

Figure 4 shows the results of this experiment: it can be seen that the training dynamics of two approaches are indeed similar, which demonstrates the viability of SWARM parallelism for heterogeneous and poorly-connected devices. We also use the T4 node preemption data of this run to demonstrate the necessity of adaptive rebalancing in a pipeline of unreliable devices; refer to Appendix H for the description.

In the next experiment, we aim to measure the pipeline throughput in different hardware conditions and to compare it with an estimate of best-case pipeline performance. We consider several setups: first, we use the same 400 preemptible T4 nodes; in another setup, we use 7 instances with 8 A100 GPU each; finally, we combine these fleets to create a heterogeneous setup. We examine the performance of the pipeline both with weight sharing and with standard, more common, Transformer blocks.

Table 3: Pipeline throughput, layer sharing.

| Hardware setup | Throughput, samples/s | | Optimal bandwidth, Mb/s | |
|---|---|---|---|---|
| | Actual | Best-case | Upload | Download |
| T4 | 17.6 | 19.2 | 317.8 | 397.9 |
| A100 | 16.9 | 25.5 | 436.1 | 545.1 |
| T4 & A100 | 27.3 | — | — | — |

Table 4: Pipeline throughput, default Transformer.

| Hardware setup | Throughput, samples/s | |
|---|---|---|
| | Actual | Best-case |
| T4 | 8.8 | 288.1 |
| A100 | 8.0 | 382.5 |
| T4 & A100 | 13.4 | — |

We measure the number of randomly generated samples processed by the pipeline both in our infrastructure and the ideal case that ignores all network-related operations (i.e., has infinite bandwidth and zero latency). The ideal case is emulated by executing a single pipeline stage 3 times locally on a single server and multiplying the single-node estimates by the number of nodes.

As demonstrated in the left two columns of Table 3 and Table 4, asynchronous training of compute-intensive models with 8-bit compressed activations regardless of the architecture specifics allows us to achieve high performance without a dedicated networking solution. Furthermore, the load balancing algorithm of SWARM allows us to dynamically and efficiently utilize different hardware without being bottlenecked by slower devices.

Next, we use the same load testing scenario to estimate the bandwidth required to fully utilize each device type in the above infrastructure. For this, we measure the average incoming and outgoing bandwidth on the nodes that serve the intermediate stage of the pipeline. We summarize our findings in the right two columns of Tables 3 and 4: it turns out that with layer sharing and 8-bit compression, medium-performance GPUs (such as T4) can be saturated even with moderate network speeds. Based on our main experiment, the optimal total bandwidth is roughly 100Mb/s higher than the values reported in Table 3 due to gradient averaging, loading state from peers, maintaining the DHT and streaming the training data. Although training over the Internet with more efficient hardware might indeed underutilize the accelerator, this issue can be offset by advanced compression strategies such as compression-aware architectures or layer sharing, as shown in Table 3.

Lastly, we evaluate the efficiency of adaptive peer rebalancing proposed in Section 3.2. We use statistics of the number of active T4 nodes from the 32-hour segment of the experiment described in the beginning of this section. We compare our strategy with a baseline that has no rebalancing and with an always optimal strategy. Appendix H contains experiment details and analysis of the results shown in Figure 5 and Table 5; notably, our strategy provides a significant improvement over the baseline that grows over time. Moreover, this improvement persists even with infrequent rebalancing.

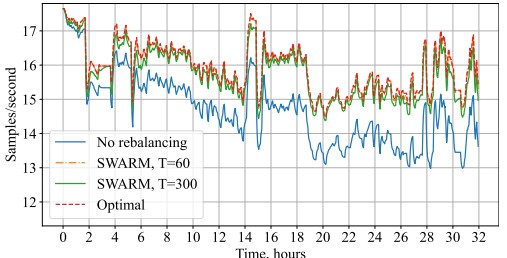

Figure 5: Throughput of rebalancing methods over time.

Table 5: Relative throughput comparison of pipeline rebalancing methods.

| Rebalancing | % of optimal | | |
|---|---|---|---|
| | Overall | First 1h | Last 1h |
| None | 82.7 | 99.0 | 45.4 |
| $T = 300$ | 95.8 | 99.4 | 88.9 |
| $T = 60$ | 97.6 | 99.8 | 91.7 |

## 5 CONCLUSION

In this work, we analyze and evaluate the feasibility of high-throughput training of billion-scale neural networks on unreliable peers with low network bandwidth. We find that this is feasible by training very large models with pipeline parallelism. To this end, we propose SWARM parallelism to overcome the challenges of pipeline parallelism for preemptible devices with heterogeneous network bandwidths and computational throughputs. We show that SWARM parallelism is highly effective at rebalancing peers and maximizing the aggregate training throughput. We also show that training **large models** with **SWARM parallelism** and **compression**-aware architectures enables high utilization of cheap preemptible instances with slow interconnect. As such, our work makes training of large models accessible to researchers that do not have access to dedicated compute infrastructure.

## REPRODUCIBILITY STATEMENT

The core contributions of our work are described either in the paper itself (including the supplementary material) or in the submitted anonymized code. We also report all key components of the experimental setup (the models that are evaluated, relevant hyperparameters and the hardware environment) in the paper. If there are specific aspects of our work with unclear reproducibility that we have not addressed during submission, we are happy to provide them during the discussion period.

## ETHICS STATEMENT

This paper studies several approaches towards making the training of large neural networks more accessible. In principle, this goal has no negative issues by itself, especially when we compare collaborative or cost-efficient training to less transparent and more expensive alternatives, such as proprietary models trained in HPC clusters. However, broader access to pretraining of large language models may have implications that need to be addressed by further research: for instance, the existence of bias and toxicity in outputs generated by such models has been observed by several independent research organizations.

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

SUPPLEMENTARY MATERIAL

## A   ANSWERS TO COMMON QUESTIONS

In this section, we provide answers to several assorted questions about our study and address some of the limitations of SWARM parallelism.

**Why not just use data parallelism with offloading?**   Regular data parallelism requires all-reduce steps where peers exchange gradients, which can be prohibitively expensive for large models. For example, a 1 billion parameter model with 16-bit gradients requires 2 GB of data to be synchronized between all $n$ devices. We need at least $n$ messages to perform this synchronization. If we have 100 devices with bidirectional communication, each client would need to send 2 GB of data to finish the synchronization. Thus, with slow interconnects, such synchronizations are not practical.

**Why not just use fully sharded data parallelism with elasticity?**   Sharded data parallelism requires all-to-all communication of parameter buffers at each layer. Each of these communications can be done in parallel and has a size of parameter count divided by $n$; in total, $n$ messages are required. Thus, for 1B parameters in 16-bit precision, a total of 2 GB need to be synchronized for both the forward and backward pass. If using low-bandwidth devices with 100 Mb/s speed, this would entail an overhead of 5.5 minutes per forward/backward pass, which is difficult to overlap with computation. This is exacerbated further, because all-to-all communication latency is determined by the slowest peer. Thus, sharded data parallelism can be particularly inefficient for setups where peers have different network bandwidths.

**ZeRO-Offload allows one to train 13B parameters on a single V100, so why do I need SWARM?**   Using ZeRO-Offload can slow down training due to the slow data transfer between external memory and the accelerator. Training with SWARM can *accelerate* training while also allowing the training of large models. See Appendix E for a more detailed comparison.

**Is it worth using preemptible instances and SWARM from an economic standpoint?**   Due to a significantly smaller cost per hour, one can leverage a larger amount of computation when using spot instances compared to on-demand cloud VMs or dedicated HPC setups. See Appendix J and Table 7 for a comparison of both hourly and total costs for an example large-scale pretraining task.

**Should I use SWARM in a supercomputer?**   By default, SWARM is worse than traditional parallelism due to extra complexity (see experiments in Section 4.2. However, SWARM can be useful in case of supercomputers that have heterogeneous devices.

**When should I avoid using SWARM?**   SWARM is very efficient at training large models with more than 1B parameters. For smaller models, a sharded data-parallel approach can be more optimal. For HPC environments with homogeneous networking, standard sharded data-parallel or pipeline-parallel training will be more efficient than SWARM because the environment is stable and predictable, so rebalancing is not required. For HPC environments which are so extensive that failure of a node is likely, the practicality of SWARM depends on how many nodes are expected to fail. Elastic sharded data parallelism is better than SWARM if the number of expected failures is relatively low.

**How much failure can SWARM handle?**   As long as there is at least one operational peer at every pipeline stage and at least one trainer, SWARM will work without any issues. The key factors defining the training run state at a given SGD step are the model parameters, the optimizer statistics, the data loader state, and the step number (required for proper scheduling). The up-to-date parameters and optimizer statistics, as well as the step number, are naturally located on all active nodes of a given stage, since they are required for training. Thus, when a peer joins the network, it can download the checkpoint corresponding to the current training state from other peers.

As we mention in Section 3.2, peer failures do not affect forward and backward passes as long as there is at least one peer at the required stage: because of rewiring, it is possible to resend activations or gradients to another worker that has identical model weights by construction. Similarly, the data loader state can be recomputed from the last known SGD step. However, we do not track the order

of examples sampled within the same microbatch; because of the IID assumption in the large-scale training setup, the distribution of gradients is expected to be the same. Hence, if the peer leaves from the pipeline stage, other workers can compute gradients and replace those accumulated by the disconnected peer, so that the number of examples for an SGD step stays the same.

**Can I use SWARM without layer sharing or quantization?** Yes, SWARM can still be effective in these scenarios. Our bandwidth experiments in the main paper give some idea what the network overhead is. By using no quantization, which means using regular 16-bit activations, the network overhead increases roughly by a factor of two. Without layer sharing, the overhead within each pipeline stage to synchronize the gradients is increased by the number of layers not being shared. As such, a rough estimate of the efficiency of SWARM in these scenarios can be estimated by taking our model size and network bandwidth requirements data and multiplying it by the relevant factor.

**How many pipeline stages can SWARM have?** While it might theoretically work with any number of pipeline stages, using long pipelines can result in reduced training throughput. Similarly to traditional pipeline parallelism, SWARM suffers from the pipeline "bubble" problem (Huang et al., 2019). More specifically, at the beginning of initial batch processing, peers near the end of the "pipeline" will be waiting for inputs. Likewise, early layers will be idle after processing the final microbatch. In theory, this can be circumvented using asynchronous updates (Narayanan et al., 2019; Yang et al., 2019), but we did not investigate them in this work due to potential convergence issues.

**Do the compression-aware architecture modifications apply only to Transformers?** Bottleneck and maxout compression are general compression techniques that can be applied to any layer in any architecture. However, their effectiveness may vary depending on where in the model they are applied and what kind of model these are applied to (for example, CNNs vs. RNNs vs. Transformers).

**Some configurations in Section 4.1 measure less than 20% GPU idle time, while many HPC systems only achieve $\approx 80\%$ GPU utilization. Does this mean that SWARM is 30% faster?** No, because these are different measurement types. Narayanan et al. (2021) measures GPU utilization as a fraction of theoretical peak FLOP/s of their GPUs. In contrast, we only measure what fraction of time the GPU is running the model, regardless of efficiency. Since no deep learning workload can achieve 100% peak FLOP/s, 20% GPU idle time for SWARM means that it can reach $\approx 0.8$x the training throughput compared to training with an infinitely fast network. As a rule of thumb, one can say that SWARM will run at a 20% slower speed than systems described by Narayanan et al. (2021) using the infrastructure that is several times cheaper.

## B  ADDITIONAL RELATED WORK

**Dynamic parameter loading.** Several recent studies propose alternative execution algorithms that allow training large models with data parallelism. Since neural networks typically use a small fraction of weights at any given moment, the remaining "inactive" parameters can be sharded (Rajbhandari et al., 2020) or offloaded to external memory (Pudipeddi et al., 2020; Ren et al., 2021; Rajbhandari et al., 2021). In sharded data parallelism Rajbhandari et al. (2020), inactive tensors are distributed across all $n$ devices such that each device stores $\frac{1}{n}$th of all parameters. For active layers, the shards are gathered such that each device holds the entire tensor just-in-time for computation. After the computation, the parameters' memory is freed so that only the sharded memory remains ($\frac{1}{n}$th per device). This makes it very memory efficient to store model and optimizer states for inactive layers if many devices are available. Similarly to tensor parallelism, these algorithms can support arbitrary models without the need for layer partitioning and can, in principle, run a large model on a single GPU, which is useful for finetuning and inference.

**Architecture-specific methods.** Finally, some distributed training algorithms take advantage of specific layers, such as locally connected layers (Dean et al., 2012; Coates et al., 2013), Mixture-of-Experts (Jacobs et al., 1991; Shazeer et al., 2017; Lepikhin et al., 2020), Switch layers (Fedus et al., 2021) or Product Key Memory (Lample et al., 2019). These layers contain many near-independent parts that can be assigned to different devices. They can easily scale to an extremely large number of parameters with a relatively small increase in compute (Shazeer et al., 2017). However, they are also less parameter-efficient (Fedus et al., 2021) and may not apply to all architectures.

**Optimal scheduling for distributed training.** When the configuration of each peer is known, it is possible to significantly optimize the pipeline scheduling by going beyond the greedy approach with global optimization techniques (Zheng et al., 2022; Tarnawski et al., 2021), even with heterogeneous hardware (Yuan et al., 2022). However, we consider a setup in which this is not possible: preemptible and volunteer peers can join at any point of the experiment, and dynamically rescheduling and orchestrating them in a centralized manner is technically difficult because of the communication and reliability constraints.

**Elastic training.** In order to train with a dynamic number of workers, deep learning practitioners have developed elastic training algorithms (TorchElastic; ElasticHorovod). If a worker leaves or fails during training, these algorithms rebalance the load between the remaining devices and continue the training procedure (Harlap et al., 2017; Ryabinin et al., 2021). If new devices join during training, they download the latest model parameters from their peers and train alongside them.

**Asynchronous training.** Another important problem is distributed training on devices with uneven performance. One way to solve this problem is to use asynchronous training, where nodes compute gradients at their own pace and aggregate them using a parameter server (Recht et al., 2011; Kijsipongse et al., 2018) or a decentralized network (Lian et al., 2017). This idea allows full utilization of each device, but may reduce the convergence rate due to "stale" gradients (Recht et al., 2011; Aji & Heafield, 2019). Several studies (Li et al., 2020; Ryabinin et al., 2021; Ren et al., 2021; Diskin et al., 2021) propose hybrid techniques that remove some synchronization points while maintaining the per-iteration convergence.

## C  STOCHASTIC WIRING DETAILS

Our approach uses *stochastic wiring*, a specialized routing algorithm designed around heterogeneous unreliable devices and high network latency. The core idea of stochastic wiring is to route each training microbatch through random devices from each pipeline stage, such that the workload of each device is proportional to its performance.

From a system design perspective, each worker runs a separate *trainer* process that forms microbatches and routes them through pipeline stages (forward and backward pass). As we describe earlier in Section 3.2, trainers run Interleaved Weighted Round Robin (Katevenis et al., 1991; Tabatabaee et al., 2020) (IWRR) scheduling to dynamically assign microbatches to peers based on each peer's training throughput ("samples per second") in a balanced way.

An important observation is that *stochastic wiring allows SWARM to mitigate network latency*. Unlike existing pipeline algorithms (Huang et al., 2019), SWARM workers do not get blocked if their neighbors take too long to process a minibatch. Instead, each SWARM device maintains a queue of microbatches assigned by trainers. In case of a latency spike, workers keep processing previously queued microbatches, maintaining high device utilization.

## D  DESCRIPTION AND COMPLEXITY OF ADAPTIVE REBALANCING

Algorithm 1 contains the formal definition of the adaptive rebalancing procedure. As described previously, each worker of SWARM that hosts model layers continuously updates the information about its load in parallel with processing the incoming requests. Each $T$ seconds, the peers measure the total load for all stages of the pipeline, and the peer with the lowest queue size from the stage with the minimum load moves to the stage with the maximum load. In principle, the algorithm could be extended to support moving multiple peers simultaneously; however, as we show in Appendix H, even in the current form the algorithm bridges most of the gap between the optimally balanced pipeline and the system without any rebalancing.

The complexity of Algorithm 1 can be estimated as follows: for $M$ as the highest number of peers over all stages, we have $O(M)$ operations in Lines 9–11 and Lines 22–24, and all other operations take constant time for a single stage. These operations are nested in the loop over all stages, which means that the total complexity of the algorithm is $O(MS)$. For practical numbers of both peers (e.g., < 10,000) and stages (fewer than 100), this incurs a negligible overhead on performance, as all communication and computation is conducted in parallel with processing the actual forward and backward passes.

Also, notice that only one migrating peer needs to stop processing requests and download the weights and optimizer statistics of the pipeline stage it starts serving: this means that the overall network load of this procedure is relatively small, as all DHT requests handle scalar data and do not exceed the number of active peers for each worker.

In practice, the algorithm handles slight deviations in local time and network/DHT latencies by allowing the peers to wait for straggling nodes in Line 9 for a predefined timeout. If a node does not join the rebalancing procedure by reporting its load in time or joins the network too late, it is omitted from the current iteration.

---

**Algorithm 1** Adaptive rebalancing for SWARM parallelism

---

**input** peer index $i$, current peer stage $s_{cur}$, total number of stages $S$, rebalancing period $T$
1: **while** active **do**
2:     Sleep for $T$ seconds
3:     Measure $q_i$ as the local request queue size
4:     Write $(i, q_i)$ as the key-subkey pair to $\text{DHT}[s_{cur}]$
5:     Initialize minimum and maximum load stages: $s_{min} = s_{max} := -1$,
6:     $l_{min} := \infty, l_{max} := -\infty$
7:     **for** $s$ in $1, \ldots, S$ **do**
8:         Initialize the load buffer $L = 0$
9:         **for** $(j, q_j)$ in $\text{DHT}[s]$ **do**
10:            $L := L + q_j$
11:         **end for**
12:         **if** $L > L_{max}$ **then**
13:            $s_{max} := s, \ L_{max} := L$
14:         **end if**
15:         **if** $L < L_{min}$ **then**
16:            $s_{min} := s, \ L_{min} := L$
17:         **end if**
18:     **end for**
19:     **if** $s_{cur} = s_{min}$ **then**
20:         // Migrate to the maximum load stage
21:         Initialize the minimum load peer $i_{min} := -1, q_{min} := \infty$
22:         **for** $(j, q_j)$ in $\text{DHT}[s]$ **do**
23:            **if** $q_j < q_{min}$ **then**
24:                $i_{min} := j, \ q_{min} := q_j$
25:            **end if**
26:         **end for**
27:         **if** $i_{min} = i$ **then**
28:            // This peer should migrate
29:            $s_{cur} := s_{max}$
30:            Download up-to-date parameters from peers in $s_{max}$
31:         **end if**
32:     **end if**
33: **end while**

---

## E   ON THE RELATION BETWEEN SWARM AND ZERO-OFFLOAD

In this section, we argue that depending on the use of DPU, SWARM-parallel training is equivalent to either fully synchronous training or the semi-synchronous training proposed in ZeRO-Offload (Ren et al., 2021). That is, SWARM produces exactly the same stepwise updates as conventional distributed training algorithms and will therefore achieve a solution in the same number of steps.

This observation is similar to how many advanced distributed training techniques (Huang et al., 2019; Rajbhandari et al., 2020) are computationally equivalent to regular synchronous training on a single device. For instance, despite using advanced distributed computation strategies, GPipe (Huang et al., 2019) computes exactly the same mathematical expression to obtain gradients and applies those gradients in the same order as any other *synchronous* training algorithm. On the other hand,

PipeDream (Narayanan et al., 2019) changes the order in which the updates are applied, introducing the so-called stale gradients (Recht et al., 2011). This allows PipeDream to improve device utilization but has been shown to reduce the final model quality in some setups (Wang et al., 2020).

Despite using randomized routing and asynchronous communication between pipeline stages, SWARM still performs optimizer steps synchronously after peers collectively reach the required global batch size (which is a hyperparameter). While different peers may accumulate a different number of samples, they will all use the same gradient after averaging. Any peer that fails or does not meet this condition is considered a straggler and must reload its state from neighbors before it can resume training. This procedure ensures that all surviving peers use non-stale aggregated gradients over the specified batch size when performing the optimizer step.

The only deviation from fully synchronous training is that SWARM uses the same approach for CPU offloading as ZeRO-Offload, and by extension, delayed parameter updates (DPU). While DPU was shown not to affect convergence (Ren et al., 2021; Stich & Karimireddy, 2020; Arjevani et al., 2020), one can disable this functionality and make SWARM fully equivalent to standard training.

Naturally, these guarantees come at the cost of reduced hardware utilization, as a small portion of devices will need to wait after every step. However, as we show in Section 4.3, SWARM can still train with competitive training throughput due to the fact that large models are trained with increased batch sizes (Brown et al., 2020).

## F    ADDITIONAL DETAILS FOR SECTION 4.1

We benchmark four versions of the Transformer layer:

- "base": $d_{model} = 768$, $d_{\text{FFN}} = 3072$, 12 heads;
- "xxlarge": $d_{model} = 4096$, $d_{\text{FFN}} = 16384$, 32 heads;
- "GPT-3" (Brown et al., 2020): $d_{model} = 12288$, $d_{\text{FFN}} = 49152$, 96 heads.
- "Ours": $d_{model} = 4096$, $d_{\text{FFN}} = 16384$, 32 heads, 3 layers per pipeline stage.

In Table 6, we report FLOP and parameter counts of each version based on the expressions from Kaplan et al. (2020). For simplicity, we set up each experiment with 12 Transformer layers using 12 servers (4 for "Ours") with a single V100-PCIE GPU each. The servers communicate at 500Mbps under 3–6ms latency.

Due to modest communication bandwidth, smaller models spend most of the time waiting for the network. However, that same bandwidth allows for $> 80\%$ GPU utilization when dealing with GPT-3-sized layers. If we co-locate 3 GPT-3 layers per pipeline stage, the GPU utilization can further improved to $> 90\%$.

The time reported in Section 4.1 is the time required to run forward and backward pass for all layers with a batch of 1x512 tokens, not including the Adam updates. All results are averaged over 1000 consecutive batches; the standard deviations are below 0.1%. All four GPUs are in the same data center but on different servers. Each layer is a `TransformerEncoderLayer` from PyTorch 1.7.0 (Paszke et al., 2019) wrapped with activation checkpointing. We use `hivemind==0.8.15` (Ryabinin & Gusev, 2020) with a single synchronous trainer based on the BERT training code from Transformers (Wolf et al., 2020). However, these results are not specific to hivemind and are likely reproducible in FairScale (Baines et al., 2021) or PyTorch RPC. The

Table 6: Parameter and FLOP counts.

| Architecture | Parameters | FLOP count |
|---|---|---|
| "base" | 7.08M | $2.2 \times 10^{10}$ |
| "xxlarge" | 201M | $6.2 \times 10^{11}$ |
| "GPT-3" | 1.81B | $5.5 \times 10^{12}$ |
| "Ours" | 201M | $1.8 \times 10^{12}$ |

only important detail is that the training code should run as much communication as possible in the background while the GPUs are busy processing batches. It is important to reuse the same connection for multiple RPC calls so that the TCP buffer does not have to warm up during each call. Also, our implementation performs quantization asynchronously with communication and other computations.

## G  ADDITIONAL DETAILS FOR SECTION 4.3

We use the standard Transformer architecture with two modifications: Rotary Positional Embeddings (Su et al., 2021) and GeGLU activations (Shazeer, 2020). Similarly to other models trained on Pile (Gao et al., 2020; Wang & Komatsuzaki, 2021), we use the tokenizer of GPT-2 (Radford et al., 2019). Following Li et al. (2021), we linearly increase training sequence length during the initial phase. More specifically, we begin training with sequences of up to 256 tokens and increase them to the maximum length of 2048 over the first $12,000$ optimizer steps. We train the model with LAMB (You et al., 2020), following the configuration from the original paper for a batch size of 16384. On top of that, we set $\eta = 10^{-3}$ and $\beta_2 = 0.95$ to account for the increased model size.

## H  ADAPTIVE REBALANCING EVALUATION DETAILS

In this experiment, we evaluate the efficiency of adaptive peer rebalancing between stages proposed in Section 3.2. We use actual statistics of the number of active T4 nodes from the 32-hour segment of the experiment described in Section 4.3 for a representative sample of training dynamics with unstable participation. We use these data to simulate training dynamics as follows: we use a sequence of events, each consisting of a timestamp and the change in the number of peers (which can be positive or negative). When a worker is removed from the pipeline, we randomly choose the stage it was removed from: that is, removing $N$ peers corresponds to $N$ samples from the uniform distribution over four pipeline stages. To compare our method with the baseline without rebalancing, we run 10 simulations over different random seeds and average the resulting trajectories.

The results of this evaluation are available in Figure 5; for reference, we also provide the performance of a theoretically optimal rebalancing strategy that maintains the highest possible throughput at every moment. It can be seen that even with the rebalancing period $T = 300$, adding dynamic rebalancing helps significantly improve the overall throughput of the pipeline. When the total number of peers is approximately stable, the rebalanced pipeline also reaches the optimal one in terms of throughput, which shows the efficiency of our strategy even when moving only one node at a time.

In addition, we observed that for some brief periods, the performance of the unbalanced pipeline exceeded the throughput of the balanced one due to random choice of disconnecting peers (dropping more from the "overrepresented" stages affects the imbalanced pipeline less). However, this held true only for $\approx 4.5\%$ of the experiment and was quickly mitigated by adaptive rebalancing.

As expected, decreasing $T$ from 300 to 60 seconds improves both the overall throughput and the speed of convergence to optimal pipeline performance. However, the effect is not as drastic compared to the increase in DHT data transfer. This is also demonstrated by Table 5, which shows the relative throughput of the three configurations compared to the optimal one. Furthermore, the table displays that although initially there is little difference between rebalancing choices, it becomes more pronounced later on as the imbalanced version "drifts further" from the optimal state.

Finally, we analyzed the scaling properties of rebalancing with respect to number of stages. To do this, we conducted experiments in the same setup as above ($T = 300$) while changing the number of pipeline stages from 4 to $\{4,\ 8,\ 16,\ 32\}$. To ensure the consistency of total throughput across all experiments, we also increased the starting number of peers accordingly while keeping the preemption rate constant. As a baseline, we also evaluate the throughput of the strategy without rebalancing.

Figure 6 shows the outcome of this experiment. As displayed in the plots, both strategies drop in performance with the increase in the stage count: while all stages should drop in performance equally in expectation, in practice, the variances are too large while the number of peers is relatively too small for the asymptotic properties to take place. This effect results in more outliers (large drops in the number of peers) in the preemption distribution for more stages. Still, rebalancing allows to partially mitigate the issue: while we observe a more consistent downward trend for the baseline strategy, the rebalanced pipeline regains its performance over time and achieves higher overall throughput.

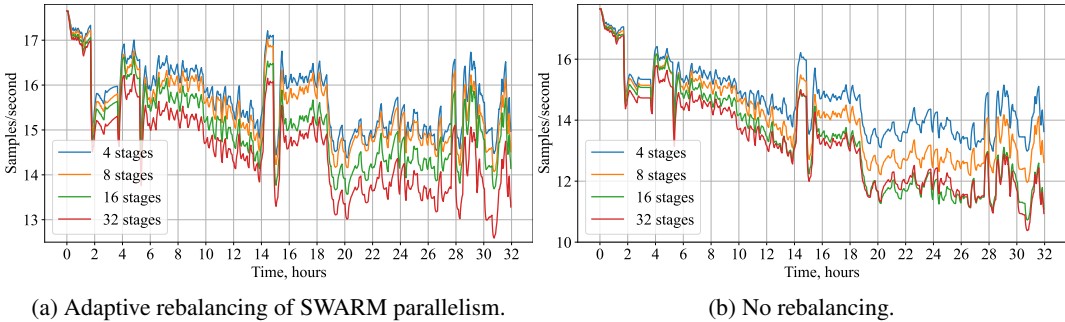

(a) Adaptive rebalancing of SWARM parallelism.  (b) No rebalancing.

Figure 6: Scaling of pipeline-parallel strategies with respect to the number of stages.

## I    ADDITIONAL SCALING EVALUATION

In this experiment, we investigate the influence of the number of nodes training with SWARM parallelism on the throughput of the pipeline. Specifically, we measure the performance of training the same model as in Section 4.3 in several configurations that differ in the size of the data-parallel group at each pipeline stage, with the number of single-GPU instances ranging from 8 to 128 (the highest quantity of preemptible nodes that we could reliably maintain for a long time). To isolate the effect of worker heterogeneity, here we use only the T4 accelerators and measure the average performance over 30 minutes of training.

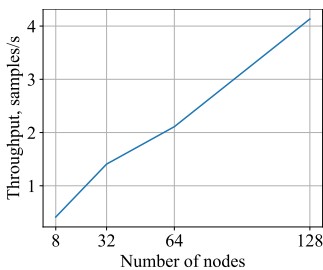

Figure 7: Scaling of SWARM throughput with the number of peers.

Figure 7 shows the results of our evaluation. It can be seen that the training performance exhibits an approximately linear scaling pattern, which can be explained by the high efficiency of both the stochastic wiring strategy and the auxiliary training components such as the DHT and the All-Reduce protocol used for gradient averaging.

## J    TIME TO SOLUTION

In this section, we evaluate the compression-aware techniques proposed in Appendix K.1 from a practitioner's point of view. A natural way to compare these techniques is in terms of "the time to solution", i.e., the wall-clock time it takes to achieve the desired validation objective. In practice, this time depends on three main factors: the compression strategy, the distributed training algorithm, and the computational infrastructure.

In order to disentangle these factors, we first address the relationship between the training algorithm and the infrastructure. As we discuss in Section 3.2 (and later in Appendix E), SWARM parallelism has the same per-iteration behavior as other synchronous methods. Theoretically, the choice of an optimal training system should come down to whichever algorithm has the highest training throughput.

To verify this argument in practice, we compare the per-iteration and per-hour performance of SWARM against fully synchronous training. For this experiment, we train the ALBERT model (Lan et al., 2020) on the WikiText-103 dataset (Merity et al., 2017). We use the ALBERT-Large architecture with 4 layer groups that correspond to 4 SWARM stages *without the architecture modifications from*

*Appendix K.1.* We follow the exact hyperparameters from the original paper: for example, we use the LAMB optimizer (You et al., 2020) with the batch size of 4096 and the sequence length of 512. We train this model in three setups: traditional distributed training with 8 V100 workers, SWARM with 8 preemptible V100 GPUs, and SWARM with 32 preemptible T4 workers.

To quantify the time to solution, we measure the wall time required to achieve the ALBERT objective equal to **1.5**. Additionally, we report the per-hour cost of each experimental setup and the total cost of achieving a loss of 1.5 using public cloud provider pricing estimates in Table 7.

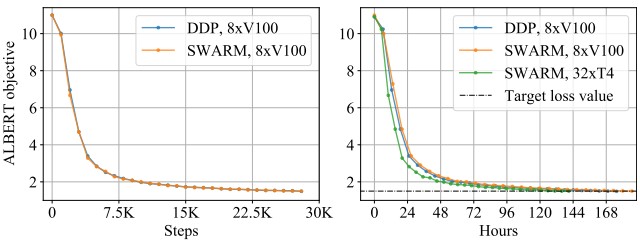

Table 7: Training time and costs.

| Setup | Time, hrs | Cost, $ | |
|---|---|---|---|
| | | Hourly | Total |
| $8 \times V100$ reliable | 175.4 | 7.834 | 1374 |
| $8 \times V100$ preemptible | 192.6 | 5.383 | 1037 |
| $32 \times T4$ preemptible | 140.8 | 3.536 | 497.8 |

Figure 8: Convergence curves of ALBERT with SWARM and standard data-parallel training.

Figure 8 (left) demonstrates that SWARM matches the per-iteration learning curves of traditional distributed training (PyTorch DistributedDataParallel) up to the variation comparable to caused by changing the random seed. However, SWARM parallelism can achieve the loss of 1.5 more cost-efficiently and faster by using preemptible instances. In turn, *when forced to use homogeneous and reliable GPUs*, SWARM would have slightly inferior performance compared to conventional algorithms, which was first demonstrated in Section 4.2.

## K COMPRESSION-AWARE ARCHITECTURES

Since pipeline parallelism has several distinct points of communication, the network overhead can be reduced considerably by reducing the size of data at these communication points. To exploit this, we develop compression-aware architectures that apply extreme compression at these points. We study two distinct communication bottleneck layers: (1) compression through a linear bottleneck layer, and (2) compression through a bottleneck induced by the maxout activation function (Goodfellow et al., 2013). We also study how compressing the activations and gradients at the communication points to 8 bits affects the predictive performance.

### K.1 DESCRIPTION

**Fully connected layers (baseline):** Fully connected layers in models such as Transformers consist of a multilayer perceptron with a single hidden layer and a nonlinear activation function. Without biases and with a residual connection (He et al., 2015) from the inputs to the outputs, this can be described as $\text{MLP}(\mathbf{x}, \mathbf{w}_1, \mathbf{w}_2) = \sigma(\mathbf{x}\mathbf{w}_1)\mathbf{w}_2 + \mathbf{x}$, where $\mathbf{x} \in \mathbb{R}^{b \times s \times m}$, $\mathbf{w}_1 \in \mathbb{R}^{m \times h}$, $\mathbf{w}_2 \in \mathbb{R}T^{h \times m}$, and $\sigma(\cdot)$ is a nonlinear activation function such as ReLU (Krizhevsky et al., 2012); $b$, $s$, $m$, and $h$ are the batch, sequence, model, and hidden dimensions of the neural network. To compress the output of the MLP layer, we want to apply a compression layer between two consecutive stages. For example, if we have 24 layers and 4 stages, we need 3 compression layers at layers 6, 12, and 18.

**Quantized activations:** A natural way to reduce the communication intensity is to send activations and gradients with respect to activations in reduced precision. However, simply casting tensors to a lower precision may slow down convergence and cause instabilities. Instead, we use dynamic 8-bit quantization with blockwise scaling from (Dettmers et al., 2021). This technique reduces communication by ≈2x and ≈4x for half and full precision, respectively.

On the other hand, quantizing and dequantizing activations can add compute overhead on every microbatch processed. Our implementation circumvents that overhead by performing quantization asynchronously on the CPU. However, this is not required, as blockwise (de)quantization takes less than 1% of total computation time: see Appendix J for details.

**Bottleneck layers:** We experiment with simple bottleneck layers that work by compressing the output features of the MLP by linear projection:

$$\text{Bottleneck}(\mathbf{x}, \mathbf{w}_1, \mathbf{w}_2, \mathbf{w}_c, \mathbf{w}_d) = \text{LayerNorm}(\text{LayerNorm}(\text{MLP}(\mathbf{x}, \mathbf{w}_1, \mathbf{w}_2))\mathbf{w}_c)\mathbf{w_d},$$

where $\mathbf{w}_c \in \mathbb{R}^{m \times c}$, $\mathbf{w}_d \in \mathbb{R}^{c \times m}$ are compression and decompression parameters with compression dimension $c < m$. We find it critical to use layer normalization Ba et al. (2016) to ensure training without divergence. The parameter matrix $\mathbf{w}_c$ resides in one stage and its outputs are transferred to the next stage that holds the parameters $\mathbf{w}_d$, which requires $m/c$ times less communication compared to the original model. Note that adding a bottleneck only adds two linear layers for the forward pass and decreases the size of MLP activations; thus, its computational overhead is negligible (less than 1% for typical sizes, see Appendix J).

**Maxout compression:** Compared to bottleneck compression, maxout compression works by using the maxout activation function (Goodfellow et al., 2013) for compression rather than a linear projection. The maxout function of factor $k$ takes inputs with a hidden dimension of $d$ and reduces this dimension by a factor of $k$ by computing the maximum value for each non-overlapping window of $k$ features. We use maxout compression as follows:

$$\text{Maxout}(\mathbf{x}, \mathbf{w}_1, \mathbf{w}_2, \mathbf{w}_d) = \text{LayerNorm}(\text{maxout}_k(\text{LayerNorm}(\text{MLP}(\mathbf{x}, \mathbf{w}_1, \mathbf{w}_2))))\mathbf{w_d},$$

where the output is reduced by a factor of $k$ through the maxout function in the previous stage, and then sent to the next stage which holds the decompression matrix $\mathbf{w}_d \in \mathbb{R}^{m/k \times m}$.

### K.2 EVALUATING THE SPEED-QUALITY TRADEOFF

While compression techniques reduce the communication overhead, they might also degrade the perplexity reached in a certain time and the final perplexity after a specific number of steps. To study these tradeoffs, we train a Transformer language model with adaptive inputs (Baevski & Auli, 2019) on the WikiText-103 dataset and measure how compression-aware architecture variants affect convergence.

Our setup follows that of (Baevski & Auli, 2019) with one difference: we use a sequence length of 2048 instead of 3072 to fit this model into our smaller GPUs. To measure the time to solution, we look at the number of iterations it takes to converge to the training perplexity of **22**. We evaluate the baseline model and three compression-aware modifications from Section K.1: bottleneck, maxout, and block-wise dynamic 8-bit quantization, each with 2 pipeline stages and each a compression factor of 2x.

Table 8: Performance of compression methods for a Transformer language model with adaptive inputs on WikiText-103. The asterisk denotes that the difference is not statistically significant. For downstream experiments, see Table 9 (Appendix K.3)

| Method | Ppl after 286K steps | Steps to ppl 22 | Data transfer | Extra compute | |
|---|---|---|---|---|---|
| | | | | Absolute | Relative |
| No compression | 21.02 | 1x | 1x | 0 | None |
| 8-bit compression | 21.13 | 0.97x* | 0.5x | 1.2ms | None (overlapped) |
| Bottleneck | 21.76 | 1.26x | 0.5x | 1.96ms | $\leq 1\%$ |
| Maxout | 21.83 | 1.28x | 0.5x | 2.04ms | $\leq 1\%$ |

The results can be seen in Table 8. We can see that 8-bit compression does not degrade the time to 22 perplexity and maintains close to the final perplexity of the baseline. The compression-aware bottleneck and maxout architectures perform equal to each other, but degrade final perplexity slightly and increase time to a perplexity of 22 by 26–28%.

Using these results, one can determine which method is optimal for their hardware setup. For instance, training with maxout with 2 pipeline stages needs 28% more steps, but accelerates the communication phase by 2x. If communication is the limiting factor, using maxout or bottleneck compression layers will offer *improved* time to perplexity despite the performance degradation. However, the same two techniques would result in slower training in a setup where network bandwidth is unlimited.

Table 9: Training of language models on the OpenWebText Corpus (OWT). The baseline model has 253M parameters and is trained for 8 GPU-days. We apply bottleneck and maxout compression to our baseline in 2 and 4 stages with a compression factor between 2–4x. WT=WikiText, PTB=Penn Treebank, 1BW=Billion word corpus.

| Model | Stages | Compression | Validation perplexity | | | | | |
|---|---|---|---|---|---|---|---|---|
| | | | OWT | LAMBADA | WT2 | WT103 | PTB | 1BW |
| Baseline | – | – | 19.7 | 86.4 | 56.2 | 35.4 | 133.0 | 80.9 |
| 8-bit Quantization | 2 | 2x | 19.6 | 89.1 | **56.0** | **35.0** | 132.7 | 79.8 |
| Bottleneck | 2 | 2x | **19.5** | 87.7 | 56.5 | 35.2 | 129.8 | 79.2 |
| Maxout | 2 | 2x | 19.6 | **85.4** | 56.6 | 35.2 | **126.8** | **78.8** |
| 8-bit Quantization | 4 | 2x | **19.7** | 87.9 | **56.3** | 35.2 | 133.9 | 79.8 |
| Bottleneck | 4 | 2x | 21.7 | 100.0 | 66.4 | 40.0 | 149.6 | 89.5 |
| Maxout | 4 | 2x | 21.4 | 89.9 | 63.9 | 39.5 | 142.1 | 86.2 |
| Bottleneck | 2 | 4x | 21.6 | 99.8 | 64.8 | 39.6 | 145.6 | 88.3 |
| Maxout | 2 | 4x | **20.5** | **89.6** | **60.0** | **37.1** | **141.7** | **83.5** |
| Bottleneck | 4 | 4x | 28.9 | 141.6 | 100.2 | 58.1 | 235.5 | 118.3 |
| Maxout | 4 | 4x | **21.3** | **93.5** | **63.6** | **39.2** | **147.7** | **89.1** |

In turn, 8-bit quantization reduces communication cost without slowing down per-iteration convergence, making it a "safe bet" for situations where the per-iteration convergence must be preserved. In our large-scale experiments (Section 4.3), we opt to using quantization since it was enough to fully saturate the GPUs. If network bandwidth is still a limiting factor, one can combine quantization with bottleneck or maxout compression to further reduce communication.

We conduct a more systematic evaluation of these compression techniques in Appendix K.3. One important observation is that, despite requiring more optimization steps, the bottleneck and maxout techniques can achieve the same or similar quality on standard LM evaluation tasks (see Table 9).

## K.3 ADDITIONAL EXPERIMENTS

The additional experiments in this section have two purposes: (1) to evaluate how compression methods vary with the number of stages and (2) to evaluate an additional setting that is closer to modern pretraining setups such as GPT-2/3.

While (1) has further implications for scaling, (2) is helpful to account for confounding factors that might have been overlooked in the main experiments on WikiText-103. The WikiText-103 baseline uses non-BPE vocabulary, a long sequence length, and uses adaptive inputs (Baevski & Auli, 2019), all of which are not frequently used in modern pretrained transformers since GPT-2 (Radford et al., 2019).

**Experimental setup:** As a baseline, we train a Transformer language model (Vaswani et al., 2017) on the OpenWebText corpus (Gokaslan & Cohen, 2019). We use the following hyperparameters: sequence size 512, 16 layers with model dimension 1024, and hidden dimension 4096 for a total of 253M parameters. We use byte pair encoding (Sennrich et al., 2016; Radford et al., 2019) with a vocabulary size of 50264 symbols. We do not use dropout or other regularization, since our models underfit. We run these experiments in Fairseq (Ott et al., 2019).

We test bottleneck and maxout compression for a compression factor of 50% and 75% compared to the original size over two and four stages. We look at how using these compression-aware architectures affects the performance compared to the compression that they achieve.

**Results:** The results of our compression-aware architectures are shown in Table 9. We can see that while the bottleneck architecture is competitive with maxout for a compression factor of 2x with two stages, maxout has better perplexities if more stages or a higher compression ratio is used. The out-of-distribution perplexities vary consistently with the in-distribution perplexity, which suggests

compression-aware architectures do not degrade the out-of-distribution performance more than the in-distribution performance. As such, the maxout compression is an effective technique to reduce the bandwidth requirements of pipeline parallel training further.

While the 8-bit blockwise quantization can only compress the activations by a factor of two (16-bit $\rightarrow$ 8-bit), it does not affect the quality as much when compared to the baseline. As such, the 8-bit quantization appears to be a reliable default choice to reduce the communication overhead for pipeline parallelism.

When considered together with the square-cube law for distributed training and SWARM parallelism, compression-aware architectures allow for better scaling of large neural networks trained over preemptible low-bandwidth peers. Thus, compression-aware architectures improve the accessibility and affordability of training large models outside HPC environments.

