# OpenReview forum: "SWARM Parallelism: Training Large Models Can Be Surprisingly Communication-Efficient"
_ICLR.cc/2023/Conference — Submitted to ICLR 2023_

### Official Review · Reviewer_8Ywq · 2022-10-25

**Confidence:** 4
**Correctness:** 3
**Technical Novelty And Significance:** 1
**Empirical Novelty And Significance:** 3
**Recommendation:** 8

**Clarity, Quality, Novelty And Reproducibility:**

The paper is well written with enough background information.
The proposed methods are novel.
I believe most experiments can be reproduced. The code is available.


**Strength And Weaknesses:**

Strength:
- This paper studies an important and unsolved problem: training LLM outside HPC clusters with heterogeneous and preemptive instances.
- The paper proposes a novel and interesting solution: SWARM parallelism, which is decentralized and highly adaptive.
- Solid experiments with throughput and convergence analysis

Weaknesses:
- This paper uses too many approximate/lossy/asynchronous methods, which can harm the accuracy.
- Stochastic wiring and adaptive rebalancing are local greedy adjustment approaches. However, I think if we know the computing speed and memory capacity of all devices, an optimal placement/routing can be computed. [1][2]
- Section 4.2 's comparison is a little bit unfair. In a typical HPC cluster, there are typically multiple GPUs on a single node connected by high-speed NVLink. Under this more realistic setting, ZeRO Offload will perform better. If you look at the results in [3], they can achieve linear scaling on HPC clusters. The authors should compare against SOTA performance on the HPC cluster when they talk about "ideal condition".

[1] Alpa: Automating Inter- and Intra-Operator Parallelism for Distributed Deep Learning, OSDI 2022
[2] Piper: Multidimensional Planner for DNN Parallelization, NeurIPS 2021
[3] Efficient Large-Scale Language Model Training on GPU Clusters Using Megatron-LM

Minor:
Missing a discussion of a recent and highly related paper: Decentralized Training of Foundation Models in Heterogeneous Environments (https://arxiv.org/abs/2206.01288)


**Summary Of The Paper:**

This paper studies the efficient training of large language models on heterogeneous and preemptive instances.
It proposes SWARM parallelism and finds that training can be made communication-efficient with SWAM parallelism.
Evaluation of a 13B model shows that the system achieves good training throughput and convergence.


**Summary Of The Review:**

This paper proposes an interesting solution to an important problem. I am leaning toward acceptance.

---

> ### Author Response · Authors · 2022-11-12
> **Author Response to Reviewer 8Ywq**
>
> Thank you for an insightful review and for your encouraging feedback! Below, we address your main concerns.
>
> > This paper uses too many approximate/lossy/asynchronous methods, which can harm the accuracy.
>
> While technically it is true, we believe that using these methods is not a direct weakness of **our work**: both compression and DPU help us achieve high system throughput, and 1) they can be disabled, as we discuss in **Appendices E and J**; 2) their impact on the quality is negligible, as we explain by references to prior work in **Appendix E** and demonstrate empirically in **Appendix K**.
>
> > if we know the computing speed and memory capacity of all devices, an optimal placement/routing can be computed. [1][2]
> > Missing a discussion of a recent and highly related paper
>
> Thank you, this recent work is certainly relevant to ours! Indeed, scheduling (especially in a heterogenous setup, as in the last paper you mentioned) could be improved **if the setup is constant and known to us beforehand**. However, we consider **a setting in which this might not be possible**: for large-scale volunteer projects, any "central" node might not have enough up-to-date information to compute an optimal placement. Furthermore, it might be challenging to ensure the integrity and optimality of such a placement if the peers are constantly joining and leaving. That said, we will include the discussion of these papers in our related work section.
>
> > Section 4.2 's comparison is a little bit unfair. In a typical HPC cluster, there are typically multiple GPUs on a single node connected by high-speed NVLink. Under this more realistic setting, ZeRO Offload will perform better. If you look at the results in [3], they can achieve linear scaling on HPC clusters. The authors should compare against SOTA performance on the HPC cluster when they talk about "ideal condition".
>
> You are correct that in HPC clusters, ZeRO would perform better; however, as we state throughout the work (**Section 1, Section 2.2, Appendix A**), our target setup is not such clusters. There are obvious overheads involved when using SWARM, and it is possible to eliminate them in case of a stable homogeneous network with fast interconnect. We put “ideal condition” in quotes precisely because we are aware of the fact that most existing large-scale training systems are designed with high-speed inter-GPU links in mind.

---

### Official Review · Reviewer_A2NV · 2022-10-26

**Confidence:** 4
**Correctness:** 2
**Technical Novelty And Significance:** 2
**Empirical Novelty And Significance:** 2
**Recommendation:** 3

**Clarity, Quality, Novelty And Reproducibility:**

I feel the work could be improved on clarity, novelty, and reproducibility. As expressed earlier, more evaluation details could be provided to help understand the results.

**Strength And Weaknesses:**

The paper addresses an important problem of making AI innovations more affordable and thus more broadly accessible. The approach of pooling together many consumer-grade, heterogeneous, and preemptible resources is a promising one. Also, using pipeline parallelism to support LLM training is a reasonable solution.

The main issue is that the paper does not provide much new insight beyond prior work. Moreover, I feel the evaluation did not effectively address important functionality and performance questions:
 1) I suspect that some form of redundancy is required to enable recovery. I did not notice any discussion of such redundancy.
 2) How does recovery mechanism handle the loss of intermediate training state such as accumulated gradients, activations (including checkpointing state), model checkpoints, data loading iterator, etc.
3)  I observe some issues in the performance results:
     (1) It is unclear whether Figure 3 is measuring forward or/and backward pass of the layer(s)
     (2) Reporting FLOP/GPU in Figure 3 would be more helpful to understand the baseline strength and claims of peak efficiency
     (3) It is unclear whether 8-bit quantization is appropriate for GPT3 training, since it is part of performance claims
     (4) The performance comparison in 4.2 is quite confusing because the size of GPT3-style model is unspecified. Also, GPipe needed
           ZeRO-Offload to fit the model even though it uses pipeline parallelism. Is SWARM more memory efficient than GPipe? It is also
          unclear whether the V100 is 16GB or 32GB because as that is important to understand why ZeRO-Offload cannot fit 4 GPT3 layers.



**Summary Of The Paper:**

The paper proposes SWARM, a system for training large language models (LLM) on alternative environments to HPC clusters that comprise of consumer-grade, preemptible, unreliable, and geographically distributed devices. SWARM employs pipeline parallelism to partition the model layers into pipeline stages on the distributed devices and coordinates the exchange of activations. SWARM uses randomized pipelines and rebalancing to handle node failures. Also, SWARM relies on the compute-bound nature of LLMs and low communication of pipeline parallelism for performance.

**Summary Of The Review:**

The central claims that (1) model scaling results in computation scaling faster than communication, and (2) pipeline parallelism communication volume is the lowest of the training parallelism techniques are both previously known. This undercuts the main contribution of the paper. Also, I feel that the functionality and performance questions above concerning whether SWARM could replace HPC clusters for LLM training were unanswered. Although, I think the paper is going in the right direction on an important problem, I feel that it is incomplete.

---

> ### Author Response · Authors · 2022-11-12
> **Author Response to Reviewer A2NV (part 2)**
>
> > (3) It is unclear whether 8-bit quantization is appropriate for GPT3 training, since it is part of performance claims
>
> We evaluate the validity of quantization for our setting in Appendix K, although at a smaller scale. Importantly, we do not quantize the entire model; we quantize only the communication buffers; in [1], this strategy was shown not to affect the predictive performance. Also, quantization-aware training of large models in general was previously demonstrated to have a marginal impact on quality even in extreme cases [2,3,4]. Finally, recent works [5] show that it is possible to apply quantization to Transformer models at the 175B scale without performance degradations.
>
> [1] 8-Bit Approximations for Parallelism in Deep Learning. Dettmers, 2015
>
> [2] Q8BERT: Quantized 8Bit BERT. Zafrir et al., 2019
>
> [3] Training with Quantization Noise for Extreme Model Compression. Fan et al., 2020
>
> [4] Understanding and Overcoming the Challenges of Efficient Transformer Quantization. Bondarenko et al., 2021
>
> [5] LLM.int8(): 8-bit Matrix Multiplication for Transformers at Scale. Dettmers et al., 2022
>
> > (4) The performance comparison in 4.2 is quite confusing because the size of GPT3-style model is unspecified
>
> **In the second paragraph of Section 4.2**, we report the size of the “GPT-3” model in terms of layer dimensions. While it is possible to give raw parameter counts, we do not think they will be as illustrative as the layer dimensions; besides, they can be found in the original GPT-3 paper. For this specific case, we have $4*(12288^2*12)\approx 7.2$ billion parameters (excluding biases and layer normalization parameters); if necessary, we can include parameter counts in the paper.
>
> > Also, GPipe needed ZeRO-Offload to fit the model even though it uses pipeline parallelism. Is SWARM more memory efficient than GPipe?
>
> SWARM is not more memory-efficient than GPipe, because it needs offloading as well. Adding ZeRO-Offload to GPipe was necessary mostly because **the original implementation** we used **did not have it**; besides, without applying offloading to both methods, it would be unfair to compare them.
>
> > It is also unclear whether the V100 is 16GB or 32GB because as that is important to understand why ZeRO-Offload cannot fit 4 GPT3 layers.
>
> Thank you for this question! Here, we used a 32GB version of V100; even the most efficient implementation would require about 120GB of RAM for optimizer statistics, master parameters and gradients. Although we did not profile the code, we hypothesize that the rest of memory allocations were because of temporary buffers used by DeepSpeed.

---

> > ### Comment · Reviewer_A2NV · 2022-11-14
> > **Response to Rebuttal (part 2)**
> >
> > 1. GPT3 training
> >
> > [5] relates to GPT3 inference which is quite different from training.
> >
> > 2. Parameter count vs. layer specs
> >
> > Memory and compute requirements are directly related to parameter count, regardless of layer specs.
> >
> > 3. Offloading
> >
> > This means offloading is a requirement for SWARM to support GPT3-sized models since even a single layer (2.4 billion parameters) will not fit into V100-32GB without offloading. I think this is an important clarification for the paper.

---

> > > ### Author Response · Authors · 2022-11-17
> > > **Author Response to Response to Rebuttal (part 2)**
> > >
> > > > [5] relates to GPT3 inference which is quite different from training.
> > >
> > > The paper does indeed focus on inference, but it reports training and finetuning experiments (albeit at a smaller scale) in appendices E and F, respectively; we apologize for not clarifying that earlier. Still, the primary justification for our performance claims is our own evaluation in Appendix K.
> > >
> > > > Memory and compute requirements are directly related to parameter count, regardless of layer specs.
> > >
> > > As per your recommendation, we will add the parameter counts to the experiment description in Section 4 to improve clarity.
> > >
> > > > This means offloading is a requirement for SWARM to support GPT3-sized models since even a single layer (2.4 billion parameters) will not fit into V100-32GB without offloading. I think this is an important clarification for the paper.
> > >
> > > To ensure we have a common perspective: the layer itself and the gradients still fit into V100-32GB even in full precision, but Adam optimizer statistics need to be offloaded.
> > > While we state that among other method details (Appendix E: "SWARM uses the same approach for CPU offloading as ZeRO-Offload"), we will clarify that in the main paper as requested.

---

> ### Author Response · Authors · 2022-11-12
> **Author Response to Reviewer A2NV (part 1)**
>
> Thank you for your review and for your detailed feedback! Please find the responses to your concerns below; if there are any questions left, we would be glad to answer them.
>
> > The main issue is that the paper does not provide much new insight beyond prior work.
> > The central claims that (1) model scaling results in computation scaling faster than communication, and (2) pipeline parallelism communication volume is the lowest of the training parallelism techniques are both previously known. This undercuts the main contribution of the paper.
>
> Our work **addresses an important problem setting and proposes a novel and practical method to solve it**; to the best of our knowledge, none of the prior large-scale training methods were designed with this kind of setup in mind, and none of them use the techniques we apply. As we mention in **point 1 of the general response**, while the square-cube law might seem obvious **to domain experts**, we believe that it is important to highlight and study it in order to demonstrate the viability of large model training over slow and unreliable networks **to the broader academic ML audience**.
>
> > I suspect that some form of redundancy is required to enable recovery. I did not notice any discussion of such redundancy.
>
> We consider two possible interpretations of redundancy: having redundant servers for the same set of model layers, or doing redundant computations for the same set of activations.
> For layer/parameter redundancy, in **Section 3.2** we state that the algorithm requires “at least one active participant per pipeline stage” and provide more details in **Appendix A**.
>
> In turn, SWARM does **not** perform redundant computations (i.e. compute the same training sample on two servers), but instead handles errors in a “lazy” manner. When a remote peer fails (and only when it does) the algorithm will seek a replacement server for a given batch of data at the same pipeline stage, as explained in the part of Section 3.2 about stochastic wiring.
>
> > How does recovery mechanism handle the loss of intermediate training state such as accumulated gradients, activations (including checkpointing state), model checkpoints, data loading iterator, etc.
>
> Thank you for the question! Activations and local accumulated gradients can be simply **recomputed by the remaining nodes**: provided that the examples come from the same distribution (and large models are usually trained on large datasets), the gradient distribution will be the same. The same is true for the data loading iterator: since it is a simple random number generator, it can be easily recomputed from the last known step. However, in the current implementation (supplementary code), we simply draw new training samples instead of the lost ones. As long as the training batch contained the same number of examples, this was enough to match the convergence rate in all our experiments.
>
> Finally, the model checkpoints (i.e. weights and optimizer states) are replicated across all nodes that hold the respective pipeline stages. Hence, as long as there is at least one node per pipeline stage, new nodes can download the checkpoint from their peers. We describe this process in more detail in Appendix C.
>
> > (1) It is unclear whether Figure 3 is measuring forward or/and backward pass of the layer(s)
>
> As we explain in **Appendix G**, we measure the time of both forward and backward passes.
>
> > (2) Reporting FLOP/GPU in Figure 3 would be more helpful to understand the baseline strength and claims of peak efficiency
>
> While it is possible to report the FLOP count for each model size given Table D.1 from the GPT-3 paper, we decided to report the processing time instead, because the **FLOP count does not show the network overhead**. If you consider that reporting FLOPs is necessary, we will add it to the figure; however, we believe that adding more information will make the figure more difficult to understand.

---

> > ### Comment · Reviewer_A2NV · 2022-11-14
> > **Response to Rebuttal (part 1)**
> >
> > Thanks for preparing detailed rebuttal to my questions.
> >
> > 1. Contributions
> >
> > The concern here is not the problem or approach, but whether square-cube law and appropriateness of pipeline parallelism are significant enough insights to be main contributions to the community. I think not.
> >
> >  2. Redundancy
> >
> > Your response broaches topics that I believe should be systematically presented and evaluated in the draft. Unfortunately, the brief discussion does not address my concerns that redundancy is not well studied by this work. For example, it seems that every parameter state needs to be duplicated, which suggests 2X hardware requirements. Also, it is unclear if the issue of activations and accumulated gradients is handled by skipping the affected samples, or something else.
> >
> >  3. FLOP/GPU
> >
> > This question was referring to achieved throughput per GPU, which is essential to understanding the efficiency of the baseline and SWARM. Processing time is not sufficient here, especially for a poor baseline.

---

> > > ### Author Response · Authors · 2022-11-17
> > > **Author Response to Response to Rebuttal (part 1)**
> > >
> > > We thank the reviewer for further comments and address them below.
> > >
> > > > The concern here is not the problem or approach, but whether square-cube law and appropriateness of pipeline parallelism are significant enough insights to be main contributions to the community. I think not.
> > >
> > > We agree that the question of the importance of the square-cube law is a subjective one and appreciate your feedback: perhaps calling it "a law" is too bold of a claim, and we are happy to tone it down if it improves your perception of the work. However, this observation and analysis is 1) not our only contribution in this work 2) important as a motivation for studying the problem of non-HPC large model training, and thus a prerequisite for the design of SWARM. In our opinion, it was necessary to discuss the communication-computation tradeoffs of larger models, because otherwise the idea of training them over the Internet would seem implausible.
> > >
> > > > Your response broaches topics that I believe should be systematically presented and evaluated in the draft. Unfortunately, the brief discussion does not address my concerns that redundancy is not well studied by this work. For example, it seems that every parameter state needs to be duplicated, which suggests 2X hardware requirements.
> > >
> > > We agree that discussing redundancy would benefit the paper. In the above example, SWARM does not incur additional hardware requirements compared to popular systems for training large models (e.g. [1]). In such systems, all optimal training configurations contain multiple pipelines, i.e., multiple nodes holding the same pipeline stage. In SWARM, the redundancy does not come from duplicating states on the same node, but from allowing existing nodes to substitute for one another. As you suggested, we will add a more detailed discussion to the corresponding FAQ entry in the nearest revision.
> > >
> > > > FLOP/GPU. Understanding the efficiency of the baseline and SWARM
> > >
> > > First, we would like to point out that Figure 3 studies the communication efficiency of different model architectures and sizes and thus does not evaluate SWARM. However, as requested, we will estimate and report these numbers in a subsequent response by the end of the review period.
> > >
> > > [1] Efficient Large-Scale Language Model Training on GPU Clusters using Megatron-LM. Narayanan et al., 2021

---

### Official Review · Reviewer_r1H3 · 2022-10-29

**Confidence:** 2
**Correctness:** 4
**Technical Novelty And Significance:** 2
**Empirical Novelty And Significance:** 2
**Recommendation:** 5

**Clarity, Quality, Novelty And Reproducibility:**

The presentation could be improved.

The method proposed in this paper sounds novel to me, however, I am not an expert in the applied side of async distributed learning.

Code to reproduce the experiments is provided. The ablation study sounds solid to me, however, I would also expect to see a trade-off between accuracy and efficiency.

**Strength And Weaknesses:**

Pros:

1. The problem is well-motivated. Indeed, training large models is important but could also be computationally expensive and time-consuming. Existing model parallelism methods propose to assign each computation device with several layers and forward/backward sequentially (the input of one device is the output of another device).
However, these methods are limited by the device memory issue and could be less efficient when the subset of layers assigned to each device requires different computation costs. Besides, each device could disconnect abruptly due to failure or preemption, which makes model parallelism more challenging.

2. The proposed method is intuitive and easy to implement. I would always encourage more on a simpler method than some complicated algorithms with massive engineering efforts and sensitivity to hyper-parameters.

3. Ablation study on training efficiency in experiment sections and appendix are quite sufficient.

Cons:

1. Although this paper is more on the empirical side of distributed learning, I am still looking forward to seeing some discussion/results on its convergence properties on the theoretical side. For example, the authors could check existing async distributed learning papers and see how they show such kind of convergence properties.

2. The main argument that the authors made on "increasing the hidden dimension will reduce the computation load per device per unit of time ..." in Section 3.1 is kind of ambiguous. According to my understanding, increasing the hidden dimension should increase both the computation cost and communication cost, we cannot conclude that ``reduce the computation load'' given the communication cost $O(n^2)$ grows slower than the computation cost $O(n^3)$.

3. The experiment sections mainly focused on time efficiency. However, I am more interested in the trade-off between efficiency and model performance. Using async randomized temporary routing for large model training is expected to degenerate the model performance and I am interested in seeing how much it will affect the model performance.

4. The presentation could be improved. For example, an experiment setup introduced could be given at the beginning of Section 4. Besides, Section 2 could be better organized to give readers a better overview of the pros and cons of the existing method. Moreover, the authors could use one paragraph to give a high-level introduction to their method in Section 1 instead of just saying "replacing traditional pipelines with randomized temporary routing between swarms of peers".

**Summary Of The Paper:**

This paper studies efficient large-scale neural network training. The authors observe that the larger the hidden dimension of models the less communication intensive. Based on such observation, the authors propose a heuristic-driven large model training method, namely SWARM.

**Summary Of The Review:**

This paper studies an important problem. The method is well-motivated and easy to implement/reproduce according to the descriptions in the paper. The ablation studies are efficient but I would also expect to see a trade-off between accuracy and efficiency. Meanwhile, for some theoretical results on convergence (similar to most distributed learning, e.g., FedAvg), the presentation could be improved and some arguments are ambiguous. Please refer to "Strength And Weaknesses" for details.

---

> ### Author Response · Authors · 2022-11-12
> **Author Response to Reviewer r1H3**
>
> Thank you for your review and a thorough evaluation of our paper! You can find the answers to your comments below.
>
> > I am still looking forward to seeing some discussion/results on its convergence properties on the theoretical side
>
> We agree that it would be helpful to have relevant theoretical results mentioned in the paper. As we discuss in **point 5 of the general response**, there are two works that study training with delayed updates and show that the effect of these delays on convergence rates is negligible. We will cite these works in an upcoming revision of the paper.
>
> > increasing the hidden dimension should increase both the computation cost and communication cost, we cannot conclude that ``reduce the computation load''
>
> You are correct that the wording here was a bit ambiguous, and we will attempt to fix it by rewriting this sentence in terms of “the fraction of time spent on communication vs computation”. Just to reiterate, we do not claim that the amount of communication decreases with increasing the hidden dimension: our aim is to spend less time on communication **compared** to the computation, as we show in Section 4.1.
>
> > The experiment sections mainly focused on time efficiency. However, I am more interested in the trade-off between efficiency and model performance. Using async randomized temporary routing for large model training is expected to degenerate the model performance and I am interested in seeing how much it will affect the model performance.
>
> For the first part of your comment: we have experiments that evaluate the quality-speed tradeoffs of different compression methods in **Appendix K**.
>
> For the second part of the comment, we would like to note that random routing **does not affect the performance**: all nodes at one stage serve the same parameters and thus yield the same outputs. As for the “async” feature, it was shown to have no noticeable impact on the results both in the original DPU paper and in the theoretical works we mentioned above.
>
> > The presentation could be improved.
>
> Thank you for these suggestions! We will definitely try to condense Section 2 and include a better description of SWARM in the introduction. However, specifying the setup at the beginning of Section 4 is difficult, because we use different settings in different experiments: one unifying aspect is the Transformer architecture, but it is already mentioned at the beginning of the section.

---

> > ### Comment · Reviewer_r1H3 · 2022-12-09
> > **Comments after AC-reviewer discussion**
> >
> > Thanks for your reply! After reading your reply and the AC-reviewer discussion, I decided to keep my current score for the following reasons:
> >
> > - **Contributions**. The method of "reducing the relative communication cost to computation cost" by increasing the hidden dimension is a solid strategy. This is also related to the novelty issue related to the "square-cube law" as raised by other reviewers.
> >
> > - **Evidence on the effectiveness of an algorithm**. The authors can either provide some theoretical results or provide rigorous experiments to evaluate the effectiveness of the algorithm. The current experiments have some flaw need to be fixed.
> >
> > - **Trade-off between efficiency and accuracy**. I would suggest the authors conduct rigorous experiments to show this instead of referring to existing papers. According to my experience, async training or sync-distributed training will all sacrifice some model accuracy due to the randomness involved during training or some data heterogeneous issue.
> >
> > Best,
> >
> > Reviewer

---

> > > ### Author Response · Authors · 2022-12-13
> > > **Response to Comments after AC-reviewer discussion**
> > >
> > > Dear Reviewer, thank you for your comment. However, we would like to ask for clarifications regarding at least two of your points: we believe that they either have already been addressed in our previous responses or are not related to the setting and key results of our work.
> > >
> > > * **Evidence on the effectiveness of an algorithm.** We kindly ask you to be more specific about "some flaws" in our current experiments: currently, such feedback does not allow us to meaningfully improve the work, which should be among the major goals of a paper review. All experiments that you requested in the original review have either been already present in the paper or are unnecessary, as we explained in our response earlier.
> > > * **Trade-off between efficiency and accuracy.** While it would be nice to have a more extensive evaluation of asynchronous training, we still believe that it is out of scope for our work: the only asynchronous part in our method is SGD with delayed steps, and we have referred to both theoretical and practical works showing little impact on quality of such an approach. Furthermore, we consider a setting in which all peers have access to the entire public dataset, and thus we do not analyze the effect of data heterogeneity: again, this is out of scope for our work.

---

### Official Review · Reviewer_rTpG · 2022-10-30

**Confidence:** 4
**Correctness:** 3
**Technical Novelty And Significance:** 3
**Empirical Novelty And Significance:** 3
**Recommendation:** 6

**Clarity, Quality, Novelty And Reproducibility:**

This paper is well-written and easy-to-understand. This is not the first time I’m reviewing this paper, but I found the current version much clearer than the old version. The paper should be able to be reproduced with the open-sourced code by any researchers and engineers with access to a typical GPU hardware. I find this paper novel, and the detailed comment can be found in the discussion in the strength above.

In terms of paper writing, I would suggest move more related works into the appendix and move more experimental results to the main paper.

**Strength And Weaknesses:**

**Strengths**

1. In general, I think the authors are working on an important and interesting new direction of distributed training. The randomized scheduling and load-balancing algorithm is novel and is insightful for the broader research community.
2. I find most experiments of the paper can show the effectiveness of the proposed SWARM parallelism method.

**Weakness**

1. The square-cube law shows that the total size of tensors to be communicated is grows at slower quadratic rate compare to the total amount of compute (flops), which grows at a faster cubic rate. However, given a fixed per-GPU memory constraint, larger models require more GPUs, which will in turn makes the total communication cost higher (because the communication cost is related to the total number of devices involved). How does the square-cube law apply to this case?
2. It would be great to include an experiment that applies SWARM parallelism to a plain Transformer model without any layer sharing/compression to purely evaluate the performance of SWARM parallelism.
3. For pipeline parallelism, how do you schedule the forward and backward passes? Do you use the GPipe schedule or the 1F1B schedule? Please elaborate.

**Summary Of The Paper:**

The paper studies the problem of training a huge deep neural network on a cluster with low interconnect bandwidth, unstable network, and preemptible nodes. The paper first shows that the computation-to-communication ratio goes up with larger models, indicating that using low-bandwidth clusters to train a huge model-parallelized model makes practical sense. Then, the paper proposes a new training algorithm named SWARM parallelism. It uses multiple devices to serve a single pipeline stage and schedules with a randomized algorithm for load-balancing and fault-tolerance. The system in addition move devices across pipeline stages to minimize the time spent on the slowest pipeline stage.

**Summary Of The Review:**

I found the paper novel and well-written. I vote for accepting the paper.

---

> ### Author Response · Authors · 2022-11-12
> **Author Response to Reviewer rTpG**
>
> Thank you for your review; we really appreciate your feedback and your observations of this version being clearer than the previous one. Below, we address the points you raised in the review:
>
> > larger models require more GPUs <…> How does the square-cube law apply to this case?
>
> This is an insightful observation; however, considering this case would complicate the explanation but ultimately lead to the same conclusion.
>
> The memory footprint of a pipeline stage depends on two main factors: persistent tensors (parameters, optimizer states) and temporary tensors (activations, gradients w.r.t. activations). As we increase the number of hidden units n, persistent tensors scale as $O(n^2)$, while temporary tensors scale as $O(n)$. This means that the memory footprint of training smaller models will be dominated by temporary tensors, while larger models will use a larger fraction of memory for parameters or Adam statistics. Larger models do indeed require more GPUs in practice, but this does not outpace the increase in compute, and overall, training larger models still becomes less network-intensive. In Section 3.1, we deliberately use a simplified compute model and ignore many real-world details (and admit to doing so) to focus on the main factors at play, later validating them in real-world settings.
>
> > It would be great to include an experiment that applies SWARM parallelism to a plain Transformer model without any layer sharing/compression to purely evaluate the performance of SWARM parallelism.
>
> Please see **point 4 of the general response**: we can include an additional comparison of throughput for such a model, but retraining a large language model from scratch during the discussion period is quite difficult.
>
> > For pipeline parallelism, how do you schedule the forward and backward passes? Do you use the GPipe schedule or the 1F1B schedule?
>
> Thank you for the question! In our experiments, we use the standard GPipe schedule: in **point 3 of the general response**, we explain the reason why 1F1B is not quite practical in our setting. We will include this brief discussion in the next revision.
>
> > move more related works into the appendix and move more experimental results to the main paper.
>
> We agree; in **point 2 of the general response**, we detail our plan of doing so. If you have different ideas on what should be moved from/to the appendix, we are happy to incorporate your suggestions.

---

> > ### Comment · Reviewer_rTpG · 2022-12-09
> > **Comments after PC discussion**
> >
> > Thanks for your reply! After reading your reply and the PC discussion meeting, I decided to keep my current score. The problem setting is interesting, but there are flaws in evaluating the proposed method.

---

> > > ### Author Response · Authors · 2022-12-13
> > > **Response to Comments after PC discussion**
> > >
> > > Dear Reviewer, we thank you for your comment. However, we would like to ask you to specify the exact flaws in evaluation that prevented you from changing the score for our work. The current justification provides little in terms of constructive feedback, which makes it difficult for us to understand the changes you require (especially because we addressed your only comment about evaluation in our updated revision).

---

### Official Review · Reviewer_fehi · 2022-10-31

**Confidence:** 3
**Correctness:** 3
**Technical Novelty And Significance:** 2
**Empirical Novelty And Significance:** 3
**Recommendation:** 8

**Clarity, Quality, Novelty And Reproducibility:**

Clarity: The paper is well written and the problems/techniques are well illustrated.
Quality: The technical designs are well motivated and sound. The experiments are strong and convincing.
Novelty: The formulated square-cube law seems to be novel and so are the design considerations. Distinctions to existing related works are clearly discussed.
Reproducibility: This might be a problem because the experimental environments are relatively complicated and implementations are not provided (would imagine the results to be hard to exactly reproduce even with implementations). But this is also understandable.



**Strength And Weaknesses:**

S1: The problem of large-scale distributed model training is important and relatively less studied.
S2: The proposed framework is based on sound observations. The technical designs are well-motivated and straightforward, with clear reasoning, explanations, and illustrations.
S3: Experimental evaluations are comprehensive and strong.

W1: Analysis and discussions on convergence are lacking.
W2: Ablation evaluations regarding the stochastic wiring and adaptive swarm rebalancing techniques seem to be missing (at least in the main content).


**Summary Of The Paper:**

This paper introduces SWARM Parallelism, a new framework for large-scale distributed training of large ML models. The framework is designed based on the generic square-cube law, and includes several designs to further handle nonrobust devices and network communications. Large-scale experiments with GPT-3 and xxlarge in comparison with state-of-the-art baselines show the efficiency advantages of the proposed framework.

**Summary Of The Review:**

Overall an interesting effort and read for the community, but direct applications are still concerning.

---

> ### Author Response · Authors · 2022-11-12
> **Author Response to Reviewer fehi**
>
> Thank you for your encouraging feedback and a great summary of our contributions! Please allow us to address your concerns below:
>
> > Analysis and discussions on convergence are lacking
>
> While we have discussed it briefly in Appendix E, we agree that this discussion could be hard to find and that more theoretical justifications might be useful. As per **point 5 of our general response**, we intend to add a small discussion of convergence with delayed updates and refer the reader to two theoretical works that cover this optimization setting.
>
> > Ablation evaluations regarding the stochastic wiring and adaptive swarm rebalancing techniques seem to be missing (at least in the main content)
>
> Indeed, the evaluation of rebalancing is contained in Appendix F due to the space constraints, and we refer to it at the end of Section 3.2 and in Section 4.3. The ablation of stochastic wiring would actually look like the third row of Table 3, but with the throughput bottlenecked by the performance of T4 nodes; moreover, excluding wiring will make the system **impractical** to use for unreliable devices (if a node disconnects before processing a response, the previous-stage node will have no destination for its outputs).
>
> > Reproducibility: This might be a problem because the experimental environments are relatively complicated and implementations are not provided
>
> We agree that setting up an environment for experiments conducted in the work is difficult. However, we’d like to point out that we provided an implementation of most key components of our work in an anonymous repository attached to the submission.

---

> > ### Comment · Reviewer_fehi · 2022-11-12
> > **Thanks for the rebuttal**
> >
> > I have read the rebuttal and it has addressed my concerns. Thanks for making the efforts.

---

### Official Review · Reviewer_uZBm · 2022-11-02

**Confidence:** 5
**Correctness:** 2
**Technical Novelty And Significance:** 2
**Empirical Novelty And Significance:** 2
**Recommendation:** 3

**Clarity, Quality, Novelty And Reproducibility:**

The “square-cube” law proposed in this paper is nice but quite obvious to me; it has been discussed and revealed in many papers such as scaling language models [1]. Also, pipeline parallelism having lower communication (in this paper’s phrasing – “pipeline parallelism naturally grows more communication-efficient with model size”) had also been revealed in many previous papers such as [2][3][4][5].
Given that the “square-cube” laws seem to be just a summarization/re-branding of several obvious facts (which have been leveraged in many papers), I don’t think the contents in section 3.1 are novel or interesting. I am not sure if this can be counted as a contribution to this paper.

[1] Scaling Laws for Neural Language Models

[2] Efficient Large-Scale Language Model Training on GPU Clusters Using Megatron-LM

[3] Alpa: Automating Inter- and Intra-Operator Parallelism for Distributed Deep Learning

[4] Piper: Multidimensional Planner for DNN Parallelization

Writing and clarity:

The writing of section 3.2 is rather ad-hoc. Is it possible to formalize a few symbols for each stage of the model and each device of the cluster, and develop an algorithm (and if possible, equations) to illustrate how exactly the stochastic wiring and rebalancing work?


Experiments:

In sec 4.2, the choices of baselines are very tricky. If the goal is to compare the performance in ideal conditions, why don’t you compare it to stronger and more practical baselines? Here are my concerns:
- Gpipe faces a peak memory issue, which limits the largest possible number of micro-batches it can use, which in turn limits its throughput because a smaller number of micro-batch causes bubbles. I think you should choose the 1F1B schedule as a baseline, which addresses this problem.
- For Zero, why don’t you choose Zero-2 or Zero-3 but Zero-offload? If the goal is to compare the performance in ideal condition, I guess Zero-2 and Zero-3 can almost always give better performance than Zero-offload (since zero-offload has offloading which would be a penalty on performance). This is also related to the cluster you choose to perform the experiments in sec 4.2 --- because I do notice that you have 7x nodes with 8 A100 each. I suppose the experiment in 4.2 should be performed on the A100 cluster (which has more GPU memory and make Zero-Offload unnecessary).


- In sec 4.3, could you elaborate on this sentence:  “with 1.01 billion parameters in total: because of layer sharing, it is equivalent to a 13B model from (Brown et al., 2020) in terms of computing requirements.” ?
I am wondering if this means the layer sharing you introduced makes your method more advantageous because it increases the compute-to-communication ratio (because this layer sharding seems to increase the flops needed per parameter). Could you perform experiments on a standard, smaller GPT-3 model without layer sharing and report the results?

- The results in Table 3 look good. But these results seem to be achieved by a combination of many techniques, including the proposed ones in this paper, and some compression techniques which can reduce communication. Is it possible to isolate these factors with an ablation study to show how much improvement your approaches exactly bring?

- Overall, I feel the experiments section should focus on how SWARM can handle heterogeneous devices, handle failures, and handle uneven bandwidth. The results presented in 4.1 is not interesting; the results presented in 4.2 is less relevant and do not contribute to many of the claims you have made earlier in the paper about SWARM.


**Strength And Weaknesses:**

Strength:

- The problem this paper is trying to address is important.
- Overall the paper is easy to follow
- The proposed technique looks sensible.


Weakness:
- In writing, the paper seems to spend too much effort explaining the background or something obvious, instead of giving an in-depth study, empirical or theoretical, about the proposed methods.

- Some contributions claimed by this paper are questionable. For example, I doubt the so-called “square-cube” law is a contribution of this paper, because It has been studied quite intensively in several large LM system papers, and has been used in many papers published before this paper. This paper seems to just re-brand this common wisdom. Accordingly, I feel the writing in section 3.1 and the experiments in 4.1 are unnecessary and does not reveal any new insight to this area.

- What remains novel in this paper is the SWARM parallelism, which is indeed interesting and trying to address a very important problem. However, I feel the authors’ execution (such as in section 3.2 and in experiments) in explaining the proposed method and proving the method is practical could have been done better. I have some detailed comments provided next.


**Summary Of The Paper:**

This paper proposes a method called “SWARM parallelism” to allow model-parallel training of large models on heterogeneous clusters (e.g., a mix of weak and strong GPUs, unreliable compute nodes or networks with uneven bandwidth).

**Summary Of The Review:**

The paper has proposed a nice and sensible idea to address the model-parallel training in a very heterogeneous cluster environment, but the paper does not provide enough evidence to prove the method is indeed practical and verify the author's many claims.

---

> ### Author Response · Authors · 2022-11-12
> **Author Response to Reviewer uZBm**
>
> Thank you for a detailed review of our paper! We address your main concerns below; please do not hesitate to ask for further clarifications in case you have any further questions.
>
> > the paper seems to spend too much effort explaining the background or something obvious
>
> > the experiments section should focus on how SWARM can handle heterogeneous devices, handle failures, and handle uneven bandwidth
>
> Thank you for your feedback! As per point 2 of our general response, we will remove parts of related work to the appendix and replace them with shortened ablations from the supplementary material. That said, while some of our background or explanations might appear obvious to an experienced reader, we believe the paper should remain accessible for readers from different backgrounds (e.g., with knowledge of modern DL, but without expertise in model parallelism).
>
> > the “square-cube” law has been studied quite intensively
>
> Please see point 1 of our general response. Briefly speaking, while there exist works that implicitly rely on this phenomenon, none of them discuss or study it in detail, especially in a non-HPC setting. To the best of our knowledge, the works given in the review focus on different problems, in some cases not even related to distributed training (scaling properties of large models **from the quality point of view**, the interplay between different parallelism schemes, **centralized** compilation of execution plans for parallel training).
>
> > I think you should choose the 1F1B schedule as a baseline
>
> Thank you for the suggestion! As explained in point 3 of the general response, it leads to performance degradations for slower networks. We can provide the comparison with such a baseline, but it is likely to underperform in our target conditions.
>
> > For Zero, why don’t you choose Zero-2 or Zero-3 but Zero-offload?
>
> The main focus of our work is training not in “ideal conditions” but in a **network of unreliable devices**. When running in these conditions, ZeRO-2 (because of reliance on ZeRO–1) and other sharding algorithms cannot recover from node failures. In other words, when a single device fails or leaves the network, the swarm will permanently lose some of the model parameters or gradients for some of the parameters.
>
> In contrast, when nodes independently store parameters in their local RAM, they do not rely on other nodes to fetch these parameters. In our experimental settings, ZeRO-2 and 3 also require at least one order of magnitude more communication between nodes, since they need to communicate model parameters (and gradients w.r.t. each parameter) as they process a given training batch.
>
> > Could you perform experiments on a standard, smaller GPT-3 model?
>
> We reply to this question in point 4 of the general response. TL;DR, specific components of SWARM were already tested on non-shared models in the appendix at the time of submission, but it is indeed possible to show a similar performance benchmark on more standard architectures, which we intend to do in the revision. It is important to note that parameter sharing makes **all** methods more advantageous: in our case, it reduces the communication load caused by activations and inter-layer gradients, and in the data-parallel case, it would reduce the load caused by parameters and parameter gradients by the same factor.
>
> > the results presented in 4.2 is less relevant and do not contribute to many of the claims you have made earlier in the paper about SWARM
>
> We believe that it is important to include these experiments in the main paper, as omitting it would raise (completely valid) questions of no empirical comparison with existing solutions.

---

### Author Response · Authors · 2022-11-12
**General Response (part 2)**

1. **[uZBm, A2NV]: novelty of the square-cube law, it has been discussed in many papers**

We respectfully disagree and argue that these works are only superficially similar. For example, [1] studies how language model quality scales with factors like the dataset size or the training compute. It does *not* consider the distributed training strategy or communication cost.

Other papers also have somewhat related observations (mostly written as side notes), but they do not affect our novelty. Notably, [2] contains an experiment (Section 5.1) where efficiency improves at scale, but 1) they simultaneously change the model size, the batch size, the number of devices, and other factors, 2) they use a hybrid framework that is only partially pipeline-parallel and 3) they have extremely high-speed interconnect. In other words, they do not study our problem and their observations do not imply the square-cube law.

To clarify, we agree that the square-cube law is “not surprising”, especially in hindsight. The reason we need this observation is to explain **why it even makes sense** to run model-parallel training in consumer-grade networks (e.g. under 200Mb/s with typical Internet latency). To the best of our knowledge, none of the mentioned works consider (and more importantly, evaluate in practice) this setup.

2. **[uZBm, rTpG]: too much effort explaining the background, add more experimental results to the main paper**

Thank you for your feedback! Indeed, some discussions of the related work (2.2 and 2.3 in particular) can be moved to Appendix B and replaced with their summaries. To fill the freed space, we can move some of the ablation studies from the appendix (for example, the ones that were possibly missed by **fehi**) to demonstrate a more thorough evaluation of SWARM.

3. **[uZBm, rTpG]: which pipeline schedule was used, 1F1B schedule instead of GPipe**

We considered 1F1B in our preliminary experiments but found a significant issue. The (interleaved) 1F1B schedule requires **two times more communication** from partitioning the model into twice as many pipeline stages. This is acceptable in high-end HPC clusters (which is where it was invented); however, in low-bandwidth networks, 1F1B is slower than the GPipe baseline that we compare against. If requested, we can report experiments with 1F1B pipelines in different settings or add the above explanation in the nearest paper revision.

4. **[uZBm, rTpG]: experiments on a plain Transformer model**

Our existing experiments in Appendix J already compare the performance of SWARM without compression to the baseline performance. However, we agree that the results of repeating the setup of Section 4.3 without layer sharing could be interesting, since many well-known pretrained models do not use such an architecture.

Section 4.3 contains two evaluations: convergence and performance on heterogeneous hardware. As we noted in the paper, the two aspects of SWARM that might potentially hurt the convergence are Delayed Parameter Updates (shown in prior work to have little effect on quality) and inter-layer compression (shown in Appendix K to not affect convergence for non-shared models in case of 8-bit quantization, which we use in our main runs). Hence, the only part where it makes sense to evaluate a non-shared model are throughput measurements: in the upcoming revision, we will provide a version of Table 3 in a smaller-scale setup due to time constraints. The expectation is that the results will stay consistent, but the overall GPU utilization will naturally decrease: this does not devalue our findings, since even in this case one might decide on a tradeoff of using less common but more communication-efficient layers.

5. **[fehi, r1H3]: analysis and discussion of convergence**

While we see the reasons that could lead to such a question, we would like to reiterate that most of the techniques we use do not affect convergence: stochastic routing sends activations between two neighboring pipeline stages; for every single stage, the parameters of all nodes **are the same**.

One part of SWARM that might affect the convergence is Delayed Parameter Updates; in Appendix E, we already argue that disabling it results in equivalence to standard training. Using delayed updates for SGD was analyzed in [3] and [4] and was directly shown to have no significant impact on the training convergence. We thank the reviewers for helping us clarify the possible concerns of readers about convergence and will include a small discussion with the above references to the appendix

[1] Scaling Laws for Neural Language Models. Kaplan et al., 2020

[2] Efficient Large-Scale Language Model Training on GPU Clusters Using Megatron-LM. Narayanan et al., 2021

[3] The Error-Feedback framework: SGD with Delayed Gradients. Stich and Karimireddy, JMLR 21 (2020)

[4] A Tight Convergence Analysis for Stochastic Gradient Descent with Delayed Updates. Arjevani et al., ALT 2020

---

### Author Response · Authors · 2022-11-12
**General Response (part 1)**

We would like to thank all reviewers for your feedback and the effort you put into reading and evaluating our paper.

We want to begin our response by pointing out that all six reviewers (even the most critical ones) unanimously agreed on the following strengths of the work:
* **The importance of the problem we study**, which is bringing large-scale distributed training outside of HPC to the benefit of research communities with fewer resources: “The problem this paper is trying to address is important” (**uZBm**), “the problem … is important is important and relatively less studied” (**fehi**), “an important and interesting new direction of distributed training” (**rTpG**), “the problem is well-motivated” (**​​r1H3**), “The paper addresses an important problem …, the approach … is a promising one” (**A2NV**), “This paper studies an important and unsolved problem” (**8Ywq**)
* **The soundness and simplicity of the method we propose** towards this goal: “The proposed technique looks sensible” (**uZBm**), “The proposed framework is based on sound observations” (**fehi**), “The randomized scheduling and load-balancing algorithm is novel and is insightful for the broader research community” (**rTpG**), “The proposed method is intuitive and easy to implement” (**​​r1H3**), “using pipeline parallelism to support LLM training is a reasonable solution” (**A2NV**), “The paper proposes a novel and interesting solution” (**8Ywq**)

We are particularly glad to see the importance of the problem acknowledged by **six independent reviewers**, because this means that the setting we study might be interesting to the research community; hence, the publication of the work can lead to a fruitful discussion given the lack of research in this field.

Below, we address the overlapping concerns of reviewers; we aim to upload a revised version of the manuscript that improves the clarity of exposition and adds requested details (including some of the experiments that were requested) next week.

---

### Author Response · Authors · 2022-11-19
**Updated Revision Comment**

Dear Reviewers, we have uploaded a new revision of the paper that takes your feedback into account. The updated parts are highlighted in blue.

Specifically,
* We have added a **high-level description of SWARM** to the **introduction**;
* We have **reduced the related work section**, either deleting less significant parts or moving them to **Appendix B**;
* We added a citation of two relevant works on the **theoretical analysis of SGD with delayed updates** to **Section 3.2**;
* We added a section about **global scheduling optimization** for distributed training to **Appendix B**;
* We reported the **FLOP and parameter counts** of models from Section 4.1 in **Appendix F**;
* We conducted an **additional experiment with the 1F1B schedule** and added its results to **Section 4.2**;
* We conducted an additional experiment with a **standard Transformer architecture** in **Section 4.3** (currently without optimal bandwidth measurements);
* Using the space freed from compressing the related work section, we moved a part of the **rebalancing ablation study** from Appendix H **to Section 4.3**.

We encourage you to read the updated paper and see if your concerns have been resolved. As you can see, no significant changes or updates to the main contributions of the paper were needed.

---

### Decision · Program_Chairs · 2023-01-20

**Decision:**

Reject

**Justification For Why Not Higher Score:**

Some concerns about the significance and originality of the "square cube" contribution and remaining concerns about experiments.

**Justification For Why Not Lower Score:**

N/A

**Metareview: Summary, Strengths And Weaknesses:**

Reviewers are in agreement about the importance of the problem studied in the paper, training large neural network models with networks of unreliable, poorly-connected devices. Even after in-person discussions, there remained division about the originality and significance of the square cube law (sec 3.1). However, there was agreement even amongst one of the most positive reviewers that the experimental evaluations are a weak point of the paper, and the author responses did not convince the reviewers otherwise, particularly around the responses to reviewer questions about the 1F1B schedule and around assumptions about the assumed impact of async updates on accuracy, and the tradeoffs achievable.

**Summary Of Ac-Reviewer Meeting:**

We identified that the key tradeoff was how positive reviewers felt about the importance of the problem, and how negative reviewers felt about weaknesses in execution and questions of significance of the square-cube law. There was little disagreement about the facts, and it came down more to a question of weighting the different factors in reaching a decision for the paper. The reviewers with lower scores ended up being more convincing, and the consensus ended up being that the weaknesses outweighed the strengths.